# How deep ocean-land coupling controls the generation of secondary microseism Love waves

Florian Le Pape [1,2✉], David Craig[1,2] & Christopher J. Bean [1,2]

Wind driven ocean wave-wave interactions produce continuous Earth vibrations at the seafloor called secondary microseisms. While the origin of associated Rayleigh waves is well understood, there is currently no quantified explanation for the existence of Love waves in the most energetic region of the microseism spectrum (3–10 s). Here, using terrestrial seismic arrays and 3D synthetic acoustic-elastic simulations combined with ocean wave hindcast data, we demonstrate that, observed from land, our general understanding of Rayleigh and Love wave microseism sources is significantly impacted by 3D propagation path effects. We show that while Rayleigh to Love wave conversions occur along the microseism path, Love waves predominantly originate from steep subsurface geological interfaces and bathymetry, directly below the ocean source that couples to the solid Earth. We conclude that, in contrast to Rayleigh waves, microseism Love waves observed on land do not directly relate to the ocean wave climate but are significantly modulated by continental margin morphologies, with a first order effect from sedimentary basins. Hence, they yield rich spatio-temporal information about ocean-land coupling in deep water.

[1] Geophysics Section, Dublin Institute for Advanced Studies, Dublin 2, Ireland. [2] iCRAG centre, University College Dublin, Dublin 4, Ireland.
✉email: flepape@cp.dias.ie

A better understanding of ocean generated seismic noise sources and associated wavefields is crucial for a variety of applications from seismic imagery[1] to subsurface monitoring[2] in addition to ocean wave climate and storm activity studies[3,4]. There are two energetic spectral windows in the microseism wavefield, the primary (dominant periods 10–20 s) and secondary (dominant periods 3–10 s). Generated in the ocean, primary microseisms originate from the direct action of propagating ocean gravity waves in shallow water. In contrast, secondary microseisms derive from second-order pressure variations, resulting from the interaction of opposing ocean wave fronts[5], leading to vertical pressurization of the ocean floor in arbitrarily deep water. The occurrence of Love waves in recorded primary microseism signals has been explained by the direct interaction between propagating ocean waves and sea-bottom topography gradients[6–8]. This mechanism is valid in shallow water for primary microseisms, but not for secondary microseisms that represent the strongest noise level in the seismic noise spectrum observed on land.

Common source locations have been observed for both Love and Rayleigh waves associated with secondary microseisms[9–11], suggesting that there may be a causal relationship. However some differences exist[12,13], including broader ranges of back azimuths observed for Love waves compared to Rayleigh waves. These have been interpreted as scattering and energy transfers from Rayleigh to Love waves controlled by sedimentary basin boundaries[9,12,13]. Bathymetry variations may also lead to P to SH conversions through scattering[14]. Love wave energy is equal to or dominant over Rayleigh waves for primary microseisms, but this is not the case for secondary microseisms with observed Rayleigh to Love wave energy ratios ranging from 0.4 to 1.2 (refs. [10,15–17]) or as low as 0.25 (ref. [18]). Furthermore, source regions for the secondary microseisms observed on land are highly debated, between sources located in deep water[19] and sources located in near-coastal or continental shelf regions[20]. Although, both cases are most likely occurring[21], secondary microseims noise sources observed from land appear to be dominant in coastal waters and continental shelf regions, where land arrays' beampower seems to correlate strongly with wave height[22,23]. Ocean wave models[24] have been used successfully to reproduce the secondary microseism energy recorded on land, outlining source variability with frequency and bathymetry, as well as seasonal variations[25–27]. Although the source locations derived from ocean wave models show broad agreement with locations determined from land-based seismic observations, some differences are observed[9,11,13]. Surface waves propagation effects related to crustal heterogeneities and wave refraction at continental margins need to be taken into account to properly understand those differences[21].

In this work, after highlighting the role that 3D path effects at continental margins play in apparent source locations of surface waves generated in deep water, we focus on the origin of secondary microseism Love waves and understanding better their relationship with Rayleigh waves. Comparing real observations with regional 3D acoustic/seismic simulations, we illustrate how continental margins influence the energy of Love waves observed on land. Finally, we discuss how our proposed mechanisms for ocean Love wave generation relate to the secondary microseism Love wave levels observed in different regions of the world.

## Results

**Ocean microseisms sources observed from land.** Due to its proximal position to the main North Atlantic low-pressure systems, Ireland is ideally located for the study of ocean microseisms. In order to locate recurrent microseism sources observed from land over a full year, seismic data recorded at two arrays (Fig. 1)

were analyzed for the period of March 2016 to March 2017. Details on the array analysis can be found in the "Methods" section. One significant advantage of these arrays is their location along the Irish coast on the North East (NE) Atlantic seaboard. There is no "contamination" along the terrestrial seismic wave propagation path from inland structures, such as sedimentary basins that could induce misleading wave propagation paths and signal amplitudes, once the microseisms have reached the shallow waters of the continental shelf.

Based on array analysis, consistent and very localized source distributions are located at or near the continental shelf over several frequency bands covering the secondary microseism spectrum. With secondary microseism sources in the North Atlantic expected to be more dominant south of Greenland and Iceland[27,28], the observations from the Irish arrays cannot be explained by ocean wave models alone, without taking into account seismic wave propagation effects. Furthermore, the common locations detected for both vertical and transverse components highlight a strong connection between both Rayleigh and Love waves as observed on land, particularly for the period band 3–5 s. Why Rayleigh and Love wave sources appear spatially well correlated is, at first, not obvious. The complexity of the NE Atlantic Irish offshore hyperextended margin[29,30] allows for both steep bathymetry variations and thick sediment basins that may, for example, lead to significant path effects on the observed microseisms locations.

**Path effects: 3D numerical simulations.** In order to reproduce and therefore better understand the land observations, we use the SPECFEM3D software package[31–33] to simulate both acoustic and seismic waves propagation in the region of interest. Two-dimensional simulation studies have shown the significant role of the water column and sediments on the secondary microseism energy, discussing the role of the continental slope acting as a strong barrier[34] and continental shelf barely affecting the ocean microseism wavefield recorded on land[35]. Here, the use of 3D enables the determination of important path effects associated with lateral heterogeneities in both bathymetry and sediments. For instance, 3D numerical simulations have demonstrated the significant role of topography effects on the wavefield associated with local earthquakes[36] and the path effects affecting the seismic signals recorded on volcanoes[37]. The regional anelastic 3D model used here is defined by the water layer (acoustic) and three viscoelastic layers characterized by sediment, crust, and mantle seismic velocities. Although the mantle layer will not have a strong influence at the seismic wave periods considered, it is important to include it as it controls changes in the crustal velocity gradients (Fig. 2). More details on the 3D regional model and associated simulations can be found in the "Methods" section.

We start by considering a single point source located 15 m below the sea surface with a source time function defined by a vertical pressure Ricker wavelet of 5 s dominant period, with a spectrum covering the secondary microseism periods (Fig. 3). In order to clearly highlight the path effect on the acoustic/seismic wavefield, the shallow acoustic source is located in a deep-water area (>3000 m) leading to a wavefield dominated by Rayleigh waves on land. The beamforming of synthetic seismic data from two arrays of synthetic land stations located in NW and SW of Ireland (representing the actual Donegal (DA) and Galley head (GA) arrays in Fig. 1a) identifies different apparent locations for the origins of body waves and surface waves (Fig. 3a). Similarly to previous work[20], the term pseudo-Rayleigh waves is used here to refer to Rayleigh surface waves traveling beneath the ocean, and therefore characterized by acoustic/elastic coupling. Whereas the

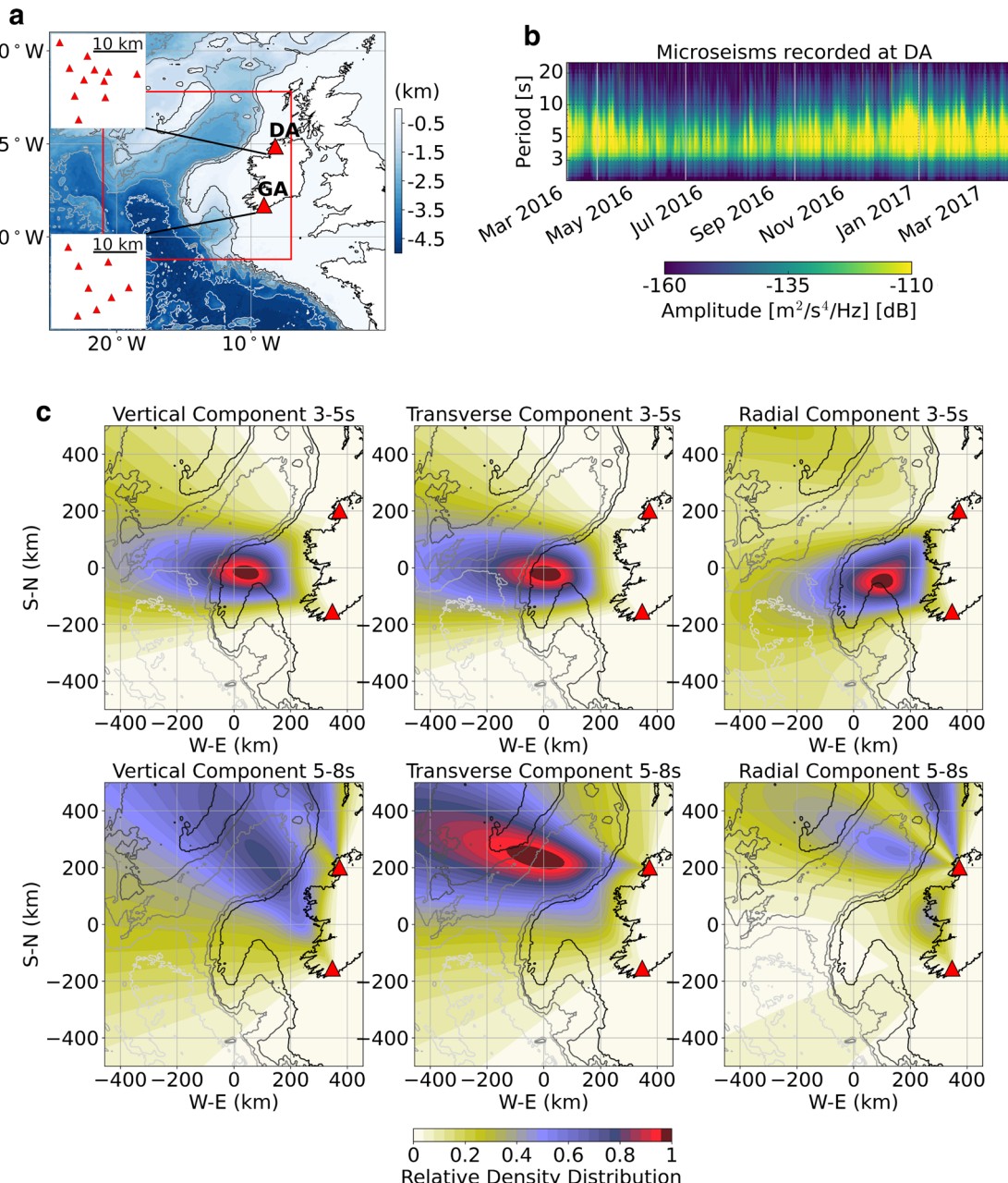

**Fig. 1 Secondary microseism sources localization from Ireland. a** Overview of the extended Irish offshore area and land seismic arrays located in Donegal (DA) and Galley head (GA), respectively, in the NW and SW of Ireland. **b** Power spectral density for a station located at array DA (Z component) for the period of March 2016 to March 2017. **c** Beamforming analysis—relative density distribution of secondary microseism sources derived from arrays DA and GA for all vertical, transverse, and radial components at period bands 3–5 and 5–8 s. The most probable source locations are characterized by the areas where the relative density distribution is maximum. Details on back azimuth, slowness and semblance for DA and GA arrays over the analysis period can be found in Supplementary Figs. 1–3. In addition, the array analysis itself and the source distribution maps generation are described in the "Methods" section.

P body waves recover broadly in the location of the original source, the recorded surface waves at the synthetic arrays mostly locate at the edge of the shelf. An apparent source location is observed, highlighting conversions from pseudo-Rayleigh, characterizing the acoustic/seismic coupling in deep water, to dominant elastic Rayleigh waves beneath shallow water (<300 m depth) for the periods of interest, combined with wave guide effects along the shelf edge (Fig. 3b). This result shows that path effects from the continental margin have a substantial impact on the apparent propagation direction of the wavefield. Furthermore, although the dominant source may be located in the deep water it

can instead appear to be generated in the shallower water near the shelf edge, as observed on the real data (Fig. 1). The simulation results are consistent with real array findings where body waves seem to mainly originate in deep water[38,39], while surface waves are more often located near the coast and shelf break[21,22,40].

**Love waves generation factors**. Rayleigh to Love waves conversions associated with continental margins have been considered previously with conversions being most significant ~20 s period[41]. However, in those calculations, the continental margin model

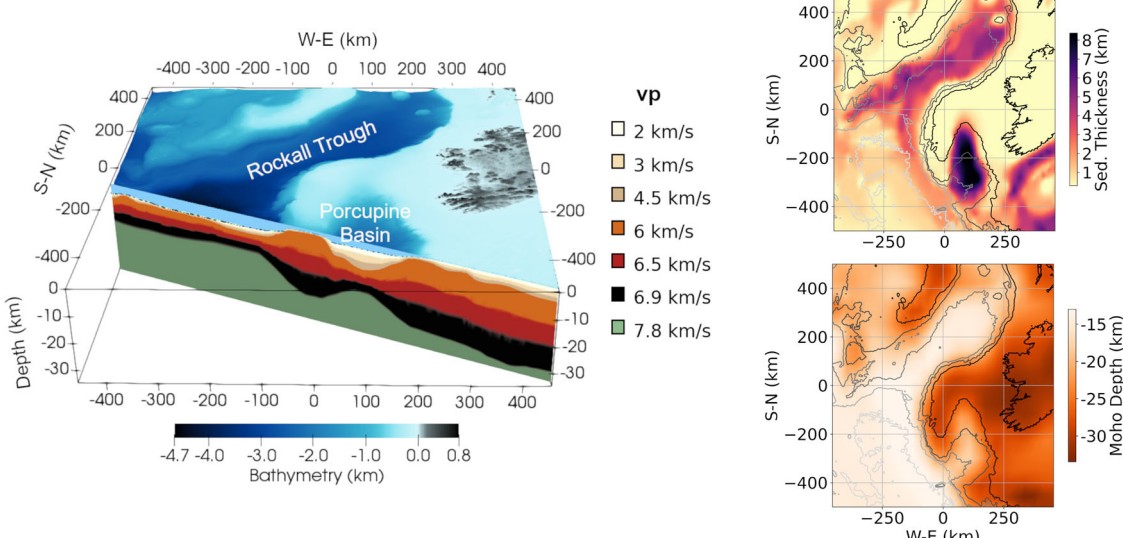

**Fig. 2 3D model used for the Irish Offshore simulations.** The model comprises most of the Irish Offshore, including thick sedimentary basins (up to 10 km in the Porcupine basin[30]). The vertical cross-section shows the $V_p$ values used for the 3D model. The model comprises water, sediments, crust, and mantle layers. The velocity gradients in the sediments and crustal layers are constrained by sediment thickness[59] and Moho depth[60]. The sediments and crustal velocity values are based on regional studies[29,30,61,62]. Shear wave velocities $V_s$, density and anelastic attenuation parameters are given in Supplementary Table 1, and the mesh quality control is described in Supplementary Fig. 4.

considered was relatively simple and therefore sediments and bathymetry variations, possibly affecting shorter periods, were not taken into account. Upper crustal heterogeneities for periods <20 s will influence transmission of Love and Rayleigh waves, as well as mode conversions occurring at the continental margin[42]. Rayleigh to Love wave conversions across strong lateral crustal heterogeneities on land have also been observed as a results of path effects on surface waves[43]. The more complex the Earth model, the more complex the radiation patterns and propagation effects. We show here that bathymetry and sediment morphologies have an unequivocal role to play. Whereas a source located on top of a flat interface will not generate transverse signals, an inclined surface will lead to horizontal shear waves generation[44]. In order to properly assess the control that heterogeneity has on seismic source generation and local offshore path effects, we first look at simple 3D simulations by taking into account bathymetry and sediment effects separately, including characteristic features of the continental margins described previously. To do so, we define a 3D concept model (Fig. 4a) whose sediment velocities, as well as both water layer and sediment thickness are progressively modified to investigate structural control over the Rayleigh and Love wavefields for a wide range of configurations. Additional details on the 3D concept model and associated simulations can be found in the "Methods" section.

The source model S1 (Fig. 4b), associated with an acoustic source on top of a bathymetry slope reveals the presence of a transverse wave front ahead of the vertical component (Z) associated with the Rayleigh wave. The transverse wavefield (T) reveals the presence of Love waves whose generation is interpreted to be associated with P to SH conversion on the bathymetry slope. It is worth noting that here the Love wave energy is relatively weak compared to the Rayleigh wave. On the contrary for the source model S2 (Fig. 4c), where the acoustic source is located above a sedimentary basin bounding edge, the seismic amplitudes recorded for all three components are comparable. Similarly to source model S1, for model S2 the Love wave generation is interpreted to be associated with SH conversion on the sedimentary basin's edge, immediately below

the source. In addition, the presence of the basin edge is likely acting as a strong SH-to-Love wave scatterer, an effect already observed for teleseismic SH waves interacting with strong bathymetric relief at ocean/continental boundaries[45].

With the introduction of sediments, the propagating wavefield becomes quite complex. For a sediment velocity, $V_p$, of 2.5 km/s, according to Supplementary Fig. 9, the Rayleigh wave is characterized by the fundamental and three higher modes with varying amplitudes, whereas the Love wave is defined only by its fundamental mode and first overtone at the considered periods. Mode conversions occurs at the edge of the sedimentary basin for both Rayleigh and Love waves, but also at the shelf (bathymetry slope) for the Rayleigh waves, as seen previously[35]. Whereas on the shelf only fundamental modes of Rayleigh and Love waves propagate, in the deep water with no sediments, Rayleigh waves are defined by the first overtone and fundamental mode due to acoustic/seismic coupling in the presence of the water layer. The wavefield animations associated with source models S1 (Supplementary Movie 2 and 3) and S2 (Supplementary Movie 4 and 5) clearly highlight that the radiation pattern of the transverse wavefield is controlled by the dip direction of the slope gradient, showing variations in Love wave energy with azimuth, whereas the energy remains more homogenously distributed for the vertical component. Furthermore, the geometry of the lateral heterogeneity defined here by the water/crust and sediment/crust interfaces will also lead to refocusing or defocusing of the wavefield, affecting its overall amplitude.

The source model S3 (Fig. 5a) highlights Rayleigh to Love wave conversions at the sediment boundary. For this model, the focus is on the deep stations in order to exclude any bathymetry effect. In this configuration, fundamental mode Love waves and first mode Rayleigh waves are dominating in the deep water. Those phases result from conversions of fundamental and higher mode Rayleigh waves excited in the sediments (Supplementary Fig. 9), and interacting with the edge of the sedimentary basin. Similarly to other models, the radiation patterns of the transverse wavefield associated with Love wave conversions (Supplementary Fig. 10) will exhibit different amplitudes, not dependent here on the

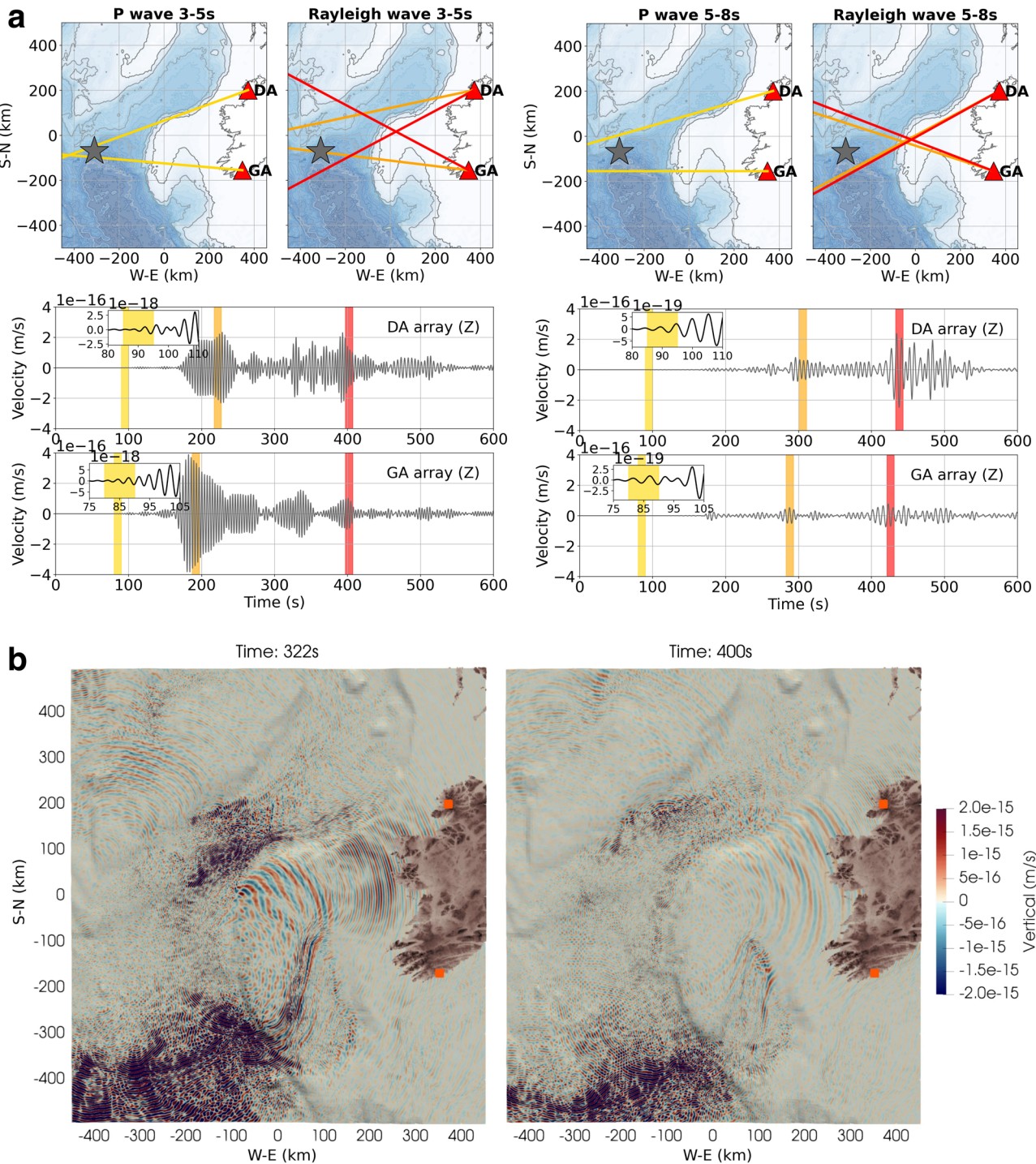

**Fig. 3 Path effects on source localization. a** Apparent source location at the shelf: synthetic array analysis for source location associated with P wave and Rayleigh wave arrivals at different period bands. The different phases of the signal recorded at the arrays do not originate at the same locations. The lines correspond to the back azimuth determined at the arrays for P wave first arrival (yellow zone and lines) and Rayleigh waves (orange, red zones and lines). Details on back azimuth, slowness, and semblance from each array can be found in Supplementary Fig. 5. Gray star indicates the actual source location for the simulation. Red triangles indicate the synthetic array locations (see Fig. 1a for detailed array geometries). The waveforms correspond to single station data recorded at each array. **b** Simulation of the vertical wavefield evolution through time as it interacts with the shelf area. Each snapshot represents the projection of the wavefield 1 km below the bathymetry. Additional snapshots and associated animation can be found in Supplementary Fig. 6 and Supplementary Movie 1. The source used for the simulations is located 15 m below the sea surface and is defined by a 5 s vertical acoustic Ricker wavelet. Orange squares define the array locations.

source location, but rather on the conversion point and its position with respect to the lateral geometry of the slope interface.

The source model S4 (Fig. 5b) focuses on acoustic–elastic pseudo-Rayleigh modes to Love wave conversions associated with the shelf break. With profile P1 perpendicularly crossing the sedimentary basin boundaries, the transverse wavefield recorded on the shelf highlights Rayleigh to Love wave conversions associated with the shelf edge (Supplementary Fig. 11). For this

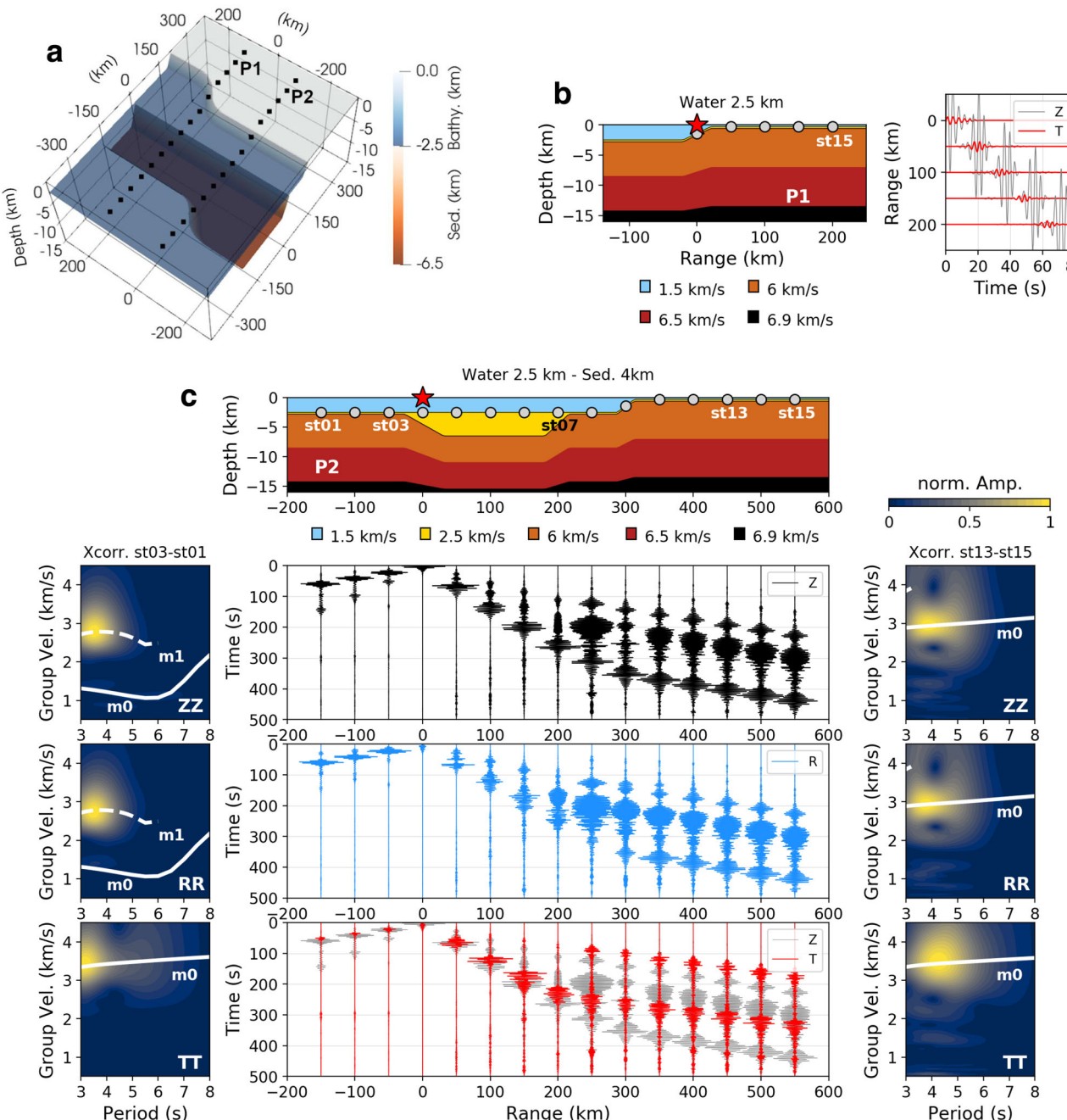

**Fig. 4 Love wave generation on top of a steep interface. a** 3D concept model: the model includes a water layer, sediments, and crust. The bathymetry slope is 5° and the slope at the edges of the sedimentary basin is 6°. The model is defined by lateral variations in bathymetry and sediment slopes, respectively, crossed by stations' profiles P1 and P2. Each profile is defined by 15 stations. In the model, the crustal interfaces are defined so that they characterize layers of equal thicknesses between the base of the sediments and 20 km depth. **b** Source model S1: source located in the water column epicentral to the center of a bathymetry slope. Cross-section of the 3D model along profile P1, as well as vertical (Z) and transverse (T) components normalized (based on vertical amplitude) seismograms for stations on the shelf, are shown. Snapshots and associated animations can be found in Supplementary Fig. 7, and Supplementary Movies 2 and 3 for both T and Z components. **c** Source model S2: source located above the slope at the edge of the sedimentary basin. Cross-section of the model along profile P2, as well as three components normalized (based on vertical amplitude) seismograms for all stations along the profile are shown. Snapshots and associated animations can be found in Supplementary Fig. 8, and Supplementary Movies 4 and 5 for both T and Z components. Due to the inline configuration of source and receivers, dispersion analysis is performed for all components through cross-correlation (Xcorr.) of pairs of stations, in order to characterize the seismic wavefield in deep water (st01–st03 pair) and on the shelf (st13–st15 pair). For comparison, theoretical dispersion curves[67] for fundamental (m0) and first overtone (m1) for Rayleigh waves (ZZ, RR) and Love waves (TT) associated with the 1D structure below each station pair are displayed. Details on the characterization of the wavefield in the sedimentary basin are described on Supplementary Fig. 9 for station st07 and different sediment velocities. The acoustic source used for all simulations is an acoustic vertical Ricker wavelet with a 5 s dominant period. All sources are located 15 m below sea surface and receivers are located 15 m below the seafloor.

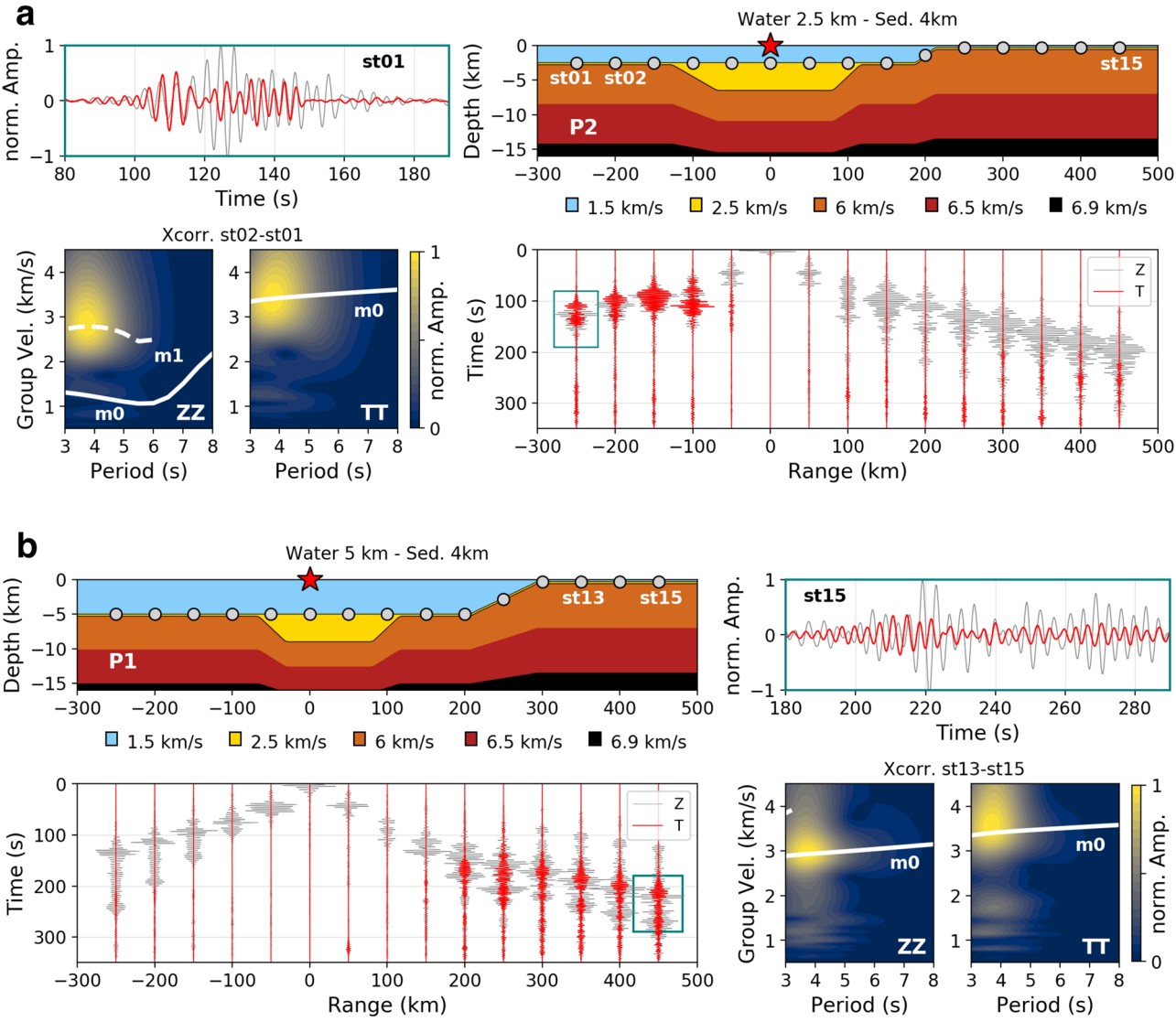

**Fig. 5 Rayleigh to Love wave conversions. a** Source model S3: source located above the middle of the sedimentary basin. Cross-section of the 3D model along profile P2, as well as vertical (Z) and transverse (T) components normalized (based on vertical amplitude) seismograms for all stations along the profile are shown. Snapshots and associated animations can be found in Supplementary Fig. 10, and Supplementary Movie 6 and 7 for both T and Z components. In this configuration, profile P2 crosses a change in the sedimentary basin lateral geometry (Fig. 4a) enabling observation of Rayleigh to Love wave conversions at the basin's edge. **b** Source model S4: source located above the middle of the sedimentary basin. Cross-section of the 3D model along profile P1 as well as Z and T components normalized (based on vertical amplitude) seismograms for all stations along the profile are shown. Snapshots and associated animations can be found in Supplementary Fig. 11, and Supplementary Movies 8 and 9 for both T and Z components. In this configuration, profile P1 is perpendicular to the sedimentary basin geometry, but crosses a change in the bathymetry lateral geometry (Fig. 4a) enabling observation of Rayleigh to Love wave conversions at the shelf area. Due to the inline configuration of source and receivers, dispersion analysis is performed for Z and T components through cross-correlation (Xcorr.) of pairs of stations, in order to characterize the seismic wavefield in deep water (st01–st03 pair) and on the shelf (st13–st15 pair). For comparison, theoretical dispersion curves[67] for fundamental (m0) and first overtone (m1) for Rayleigh waves (ZZ) and Love waves (TT) associated with the 1D structure below each station pair are displayed.

model, the Rayleigh wavefield is dominated by higher modes in the sedimentary basin and deep water (mode characterization shown in Supplementary Fig. 12), whose conversions at the shelf break result in both Rayleigh and Love fundamental modes. In contrast, in a model configuration where the Rayleigh fundamental mode is clearer in the deep water with limited higher modes (Supplementary Fig. 13), the pseudo-Rayleigh waves fundamental mode seems to convert to apparent Rayleigh waves seen on the transverse component. This feature shows that the pronounced acoustic/elastic coupling of the Rayleigh fundamental mode leads to scattering of Rayleigh waves into the transverse component at the shelf edge, although relatively weak in

amplitude. In contrast, the first overtone interaction with the shelf generates a faster phase on the transverse component consistent with Love wave velocities (Fig. 5b and Supplementary Fig. 13).

Although the Love wave generation mechanisms discussed above were previously proposed as possible candidates[12,14,16], we aim to be quantitative by searching for evidence that they can lead to the L/R energy ratios broadly observed on land[10,15,16,18]. To do so, we discuss the structural control over Love to Rayleigh kinetic energy wave (L/R) ratios observed from two separate stations located on the shelf (Fig. 6) by varying water column and sediment thickness as well as sediment velocity in the 3D concept

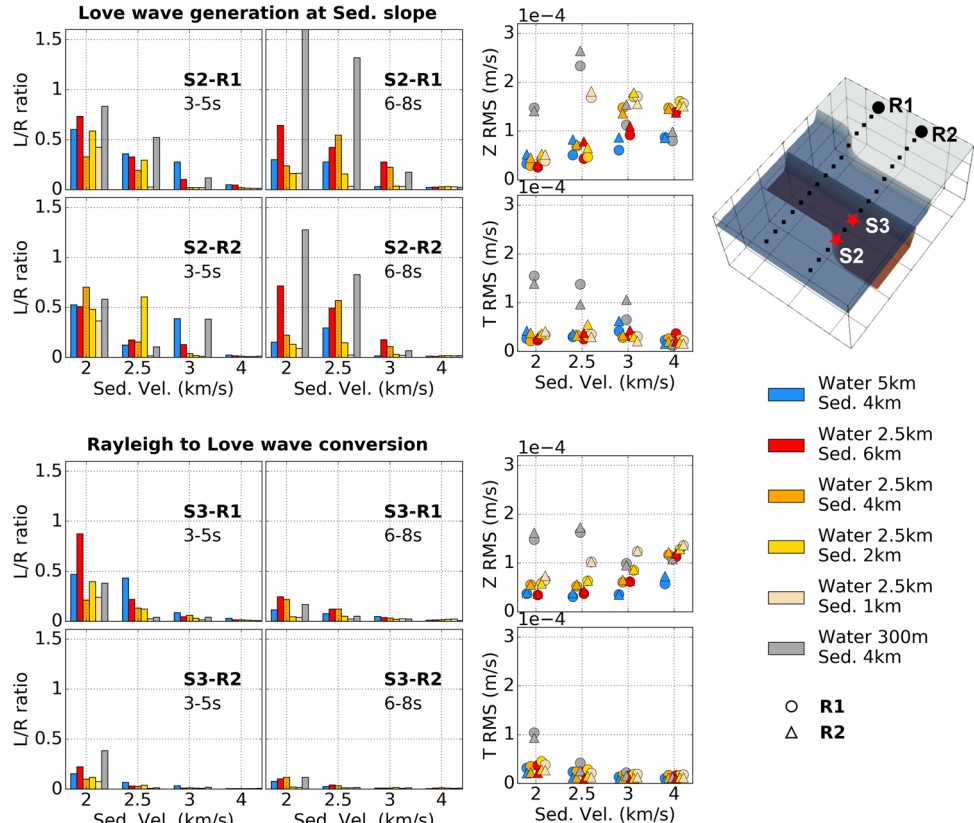

**Fig. 6 Structural control over Love to Rayleigh (L/R) kinetic energy ratio.** The ratios are estimated here for two configurations: Love wave generated at the source (source model S2 of Fig. 4c) and Rayleigh to Love waves conversions (source model S3 of Fig. 5a). For each model, different water and sediment thickness, as well as sediment velocity ($V_p$ = 2, 2.5, 3, and 4 km/s) are considered. In order to look at azimuthal changes in the ratios, the L/R ratios are estimated at receivers R1 and R2 located on the shelf, and corresponding to the stations st15 of both profiles P1 and P2 (Fig. 4). Details on the ratio estimations can be found in the "Methods" section and Supplementary Figs. 18 and 19. In addition, in order to understand the overall contribution of both Rayleigh and Love wavefields on the ratios, the RMS amplitude of vertical and transverse components (filtered in the band 3–8 s) are compared. For the RMS amplitude plots, since R1 and R2 receivers are not at the exact same distance to the sources, geometrical spreading correction was applied.

model (Fig. 4a). The calculation of L/R ratios is described in the "Methods" section. Since the ratios only represent a relative comparison between Love and Rayleigh wave energy, we also look at the RMS amplitude for associate transverse and vertical components, and how they vary with the geological environment. In addition, in order to get an idea of the water column and control of sediment structure over both Rayleigh and Love wavefields recorded at the seafloor, the Love and Rayleigh wavefields associated with each model have been investigated in detail. The results are shown in the Supplementary Figs. 9, 12, 14–17 through dispersion analysis of the three-component seismic signal recorded at station st07 in the sedimentary basin for source model S2 (Fig. 4c).

Overall, looking at vertical and transverse RMS amplitudes (Fig. 6), the dominant factor controlling the L/R ratios appears not to be Love wave energy, but instead how Rayleigh wave amplitudes fluctuate with the changing ocean depth and subsurface environment, showing in particular a high dependence on the sediment velocities at the seafloor. In fact the observed damping effect of the sediments on the Rayleigh wavefield is in agreement with other studies[12,35]. Here, we show that even a 1 km sedimentary layer, if associated with very slow sediment velocities, dampens the Rayleigh wavefield leading to relatively high L/R ratios on the shelf (Fig. 6). The highest L/R ratios are obtained for a source located above a sedimentary basin bounding edge for both short and long period secondary microseisms, making it a dominant mechanism for the Love wave contribution.

Rayleigh to Love wave conversions can lead to relatively high transverse amplitudes in the propagation direction, as seen in Supplementary Fig. 10 and 11, but for this mechanism the amount of Love waves, with respect to Rayleigh waves, remains limited due to the dominant forward scattering of the fundamental mode Rayleigh waves at the interface[46]. The nature of Love wave radiation patterns will also affect the observed ratios as a function of the source-receiver back azimuth, since bathymetry and sediment morphologies modulate both wavefields radiation behaviors, as previously suggested[10] and shown in Supplementary Figs. 7, 8, 10, and 11. For the source model S2, both Z and T components exhibit azimuthal variations, but for the Rayleigh to Love wave conversions the resulting changes in the L/R ratios seem to be more controlled by Love waves (Fig. 6).

Another key factor appears to be the water column. For instance, the shallow water model (300 m) highlights different trends in T and Z RMS amplitudes compared to deeper water models, but also high L/R ratios particularly for the period band 6–8 s (Fig. 6). The presence of sediments in deep water creates a very complex Rayleigh wavefield with increasing dominance of higher modes as sediment velocities decreases (Fig. 7 and Supplementary Fig. 9, 12, 14–17), which will affect the Rayleigh waves amplitudes observed on land. In fact, similar observations can be made for Love waves. Whereas theoretically Love waves excited in deep or shallow water should be independent of the water layer thickness, Love waves generated with a source on top of the sedimentary basin's edge exhibit differences in the modes

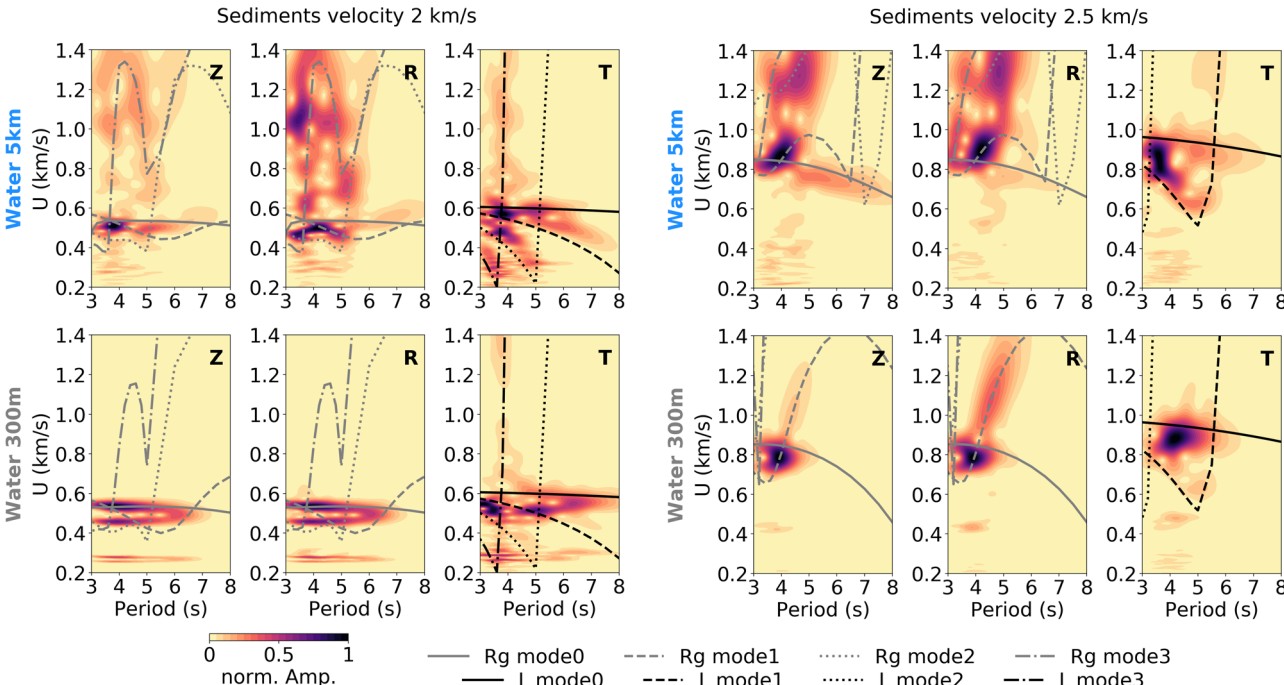

**Fig. 7 Water column effect on the Love wavefield.** Dispersion analysis for data extracted in the sedimentary basin at station st07 for the source model S2 configuration (Fig. 4c) with varying water column, sediment thickness, and velocity. On each plot the amplitude is normalized over all periods in order to see phases of higher energy. Group velocity ($U$) against period is shown for all three components: vertical (Z), radial (R) and transverse (T), and for varying sediments velocities. Theoretical dispersion curves for the associated 1D model at station st07 are computed using ref. [67]. They are shown for the first four modes of Rayleigh (gray lines) and Love waves (black lines). The slight shift in group velocities between theoretical curves and simulations are due to the 3D structure of the basin with the source located above the sedimentary basin's edge.

excited, depending on the water depth (Fig. 7). Love wave higher modes seem to dominate the wavefield at longer periods when generated in deep water. On the contrary, in shallow water, the generated Love waves show a stronger dominance of the fundamental mode. One possible explanation is that SH conversions and scattering at the basin's edge will likely exhibit a similar ocean site effect as observed for P and SV waves[47], but also for Rayleigh waves[48].

Finally, with a source located epicentral to the sedimentary basin's edge, an increasing slope will increase the transverse amplitudes recorded on the shelf, but decrease the vertical amplitudes affected by transmission across the basin (Supplementary Fig. 20). In fact, for the shallow water model, with most of the fundamental mode Rayleigh wavefield located in the sediments, the relatively low RMS vertical amplitudes observed on the shelf for a model with a sediment velocity of 2 km/s (Fig. 6), shows that a good portion of the wavefield remains trapped in the basin due to the high velocity contrast at the sedimentary basin's boundary. However, it is worth noting that taking into account attenuation in the simulations shows that, with most of the wavefield located in the sediments, amplitudes recorded in shallow water are more strongly attenuated than for deeper water models (Supplementary Fig. 20). In fact, overall vertical amplitudes are also more affected by anelastic attenuation than transverse amplitudes which leads to higher L/R ratios in more attenuative environments.

**How continental margins influence the energy of Love waves.** The energy ratios of Love waves over Rayleigh waves observed on land will reflect the balance between the microseism source intensity distribution between deep and shallow water, as well as effects from the underlying bathymetry and subsurface structure

on the generated seismic wavefield, particularly on Rayleigh waves. In order to see how all those environmental factors might lead to the observed Love wave levels in field seismic data, we now go back to our anelastic more complex 3D regional synthetic model (Fig. 2) and consider an "extended source" (Fig. 8a). The source is based on the microseism source excitation (P2L) data derived from ocean wave models and representing the surface equivalent pressure resulting from nonlinear ocean wave–wave interactions[24]. Here, we calculate the annual median of P2L data for the region as a source grid for the 3D numerical simulation. Through power spectrum interpolation, the P2L spectrum at each grid point is transformed into a continuous acoustic signal with random phase. This leads to a multipoint spatially distributed acoustic source, continuous in time and containing the full spectrum of the secondary microseism excitation over the area defined by the acoustic/elastic Earth 3D model. More details on the P2L source can be found in the "Methods" section. Synthetic microseism "observed" source location maps are then derived from array analysis of the synthetic DA and GA array seismograms generated in this "extended source" 3D simulation (Fig. 8b). Whereas the strongest ocean source area is located further than 600 km offshore to the NW (Fig. 8a), the beamforming results from combined synthetic arrays DA and GA reveal a much more focused source localization, in strong agreement with array analysis results for observed field data from actual DA and GA arrays (Fig. 8b). As discussed previously, the observed localization for Rayleigh surface waves reveals apparent source locations influenced by conversions and wave guide effects along the shelf edge. Although the transverse signal will be similarly affected, as shown above, the recorded transverse signal on land will also be highly dependent on the orientation of the sediment-filled basin edges that controls the signal radiation toward the arrays. For the NE Atlantic secondary microseisms

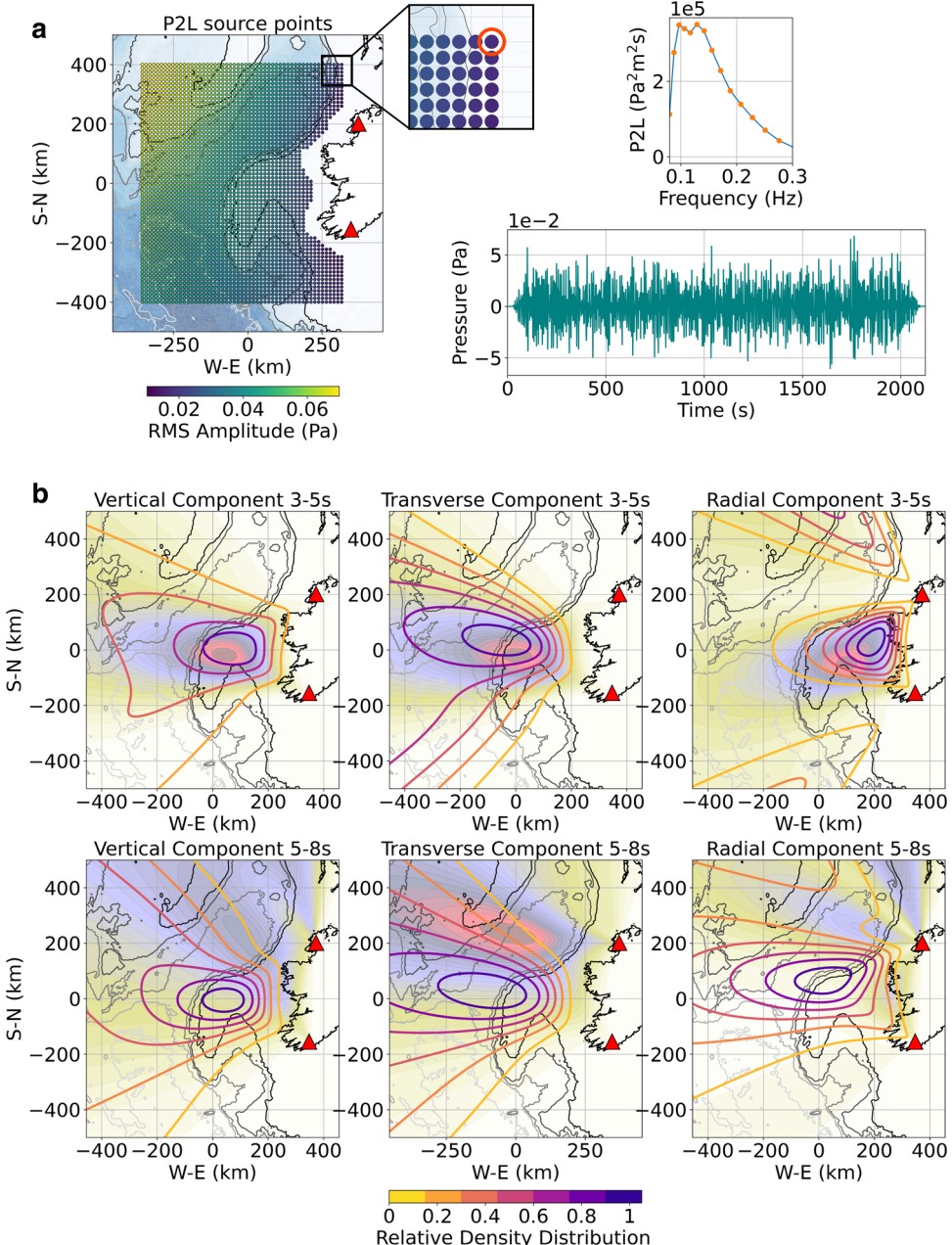

**Fig. 8 Regional 3D acoustic/seismic simulation. a** Source distribution: the simulation source is based on secondary microseism excitation or ocean surface equivalent pressure data derived from ocean wave simulations (P2L data available through Ifremer ftp://ftp.ifremer.fr). The P2L data represent the power spectral density (PSD) of pressure as a function of latitude and longitude at time intervals of 3 h. The P2L frequencies are defined by the secondary microseism frequencies, which are double the frequencies of ocean waves. Here, we take the median of the P2L data over a year (March 2016–March 2017) to generate a median P2L frequency spectrum for each geographical point, discretized here as a source grid for the simulation. Each single point source, as shown on the picture, is then defined by a random phase pressure signal derived through interpolation of their associated P2L spectrum (orange dots). Each generated pressure signal is injected into the simulation defining a multipoint continuous acoustic source throughout the whole duration of the 3D simulation and which covers most of the area defined by the model. Each point source is located 15 m below sea surface and characterized as a vertical pressure source. More details can be found in the "Methods" section. **b** Synthetic array analysis: the analysis has been performed for vertical, transverse, and radial components at period band 3–5 and 5–8 s from seismic signal generated through the numerical simulation. The locations of the seismic arrays are the same as for the observed data analysis and are represented by the red triangles. The synthetic maps (line contour maps) overlie the annual averaged source locations derived from the analysis of actual field array data (see also Fig. 1c). Details on back azimuth, slowness, and semblance for the synthetic DA and GA arrays analysis can be found in Supplementary Figs. 21–23.

sources, both the DA and GA arrays show relatively consistent back azimuths through time for the 3–5 s period band, particularly for array GA and without any significant seasonal variation (Supplementary Figs. 1–3). As highlighted previously, such consistency is dictated by the bathymetry, sedimentary basins, and

morphologies of the shelf edge, whose influence seems to dominate over the actual distribution of the microseism sources. Over the 5–8 s band, the transverse wavefield remains in agreement with observations, confirming Love wave generation at the continental margin for this period band too. However, the more

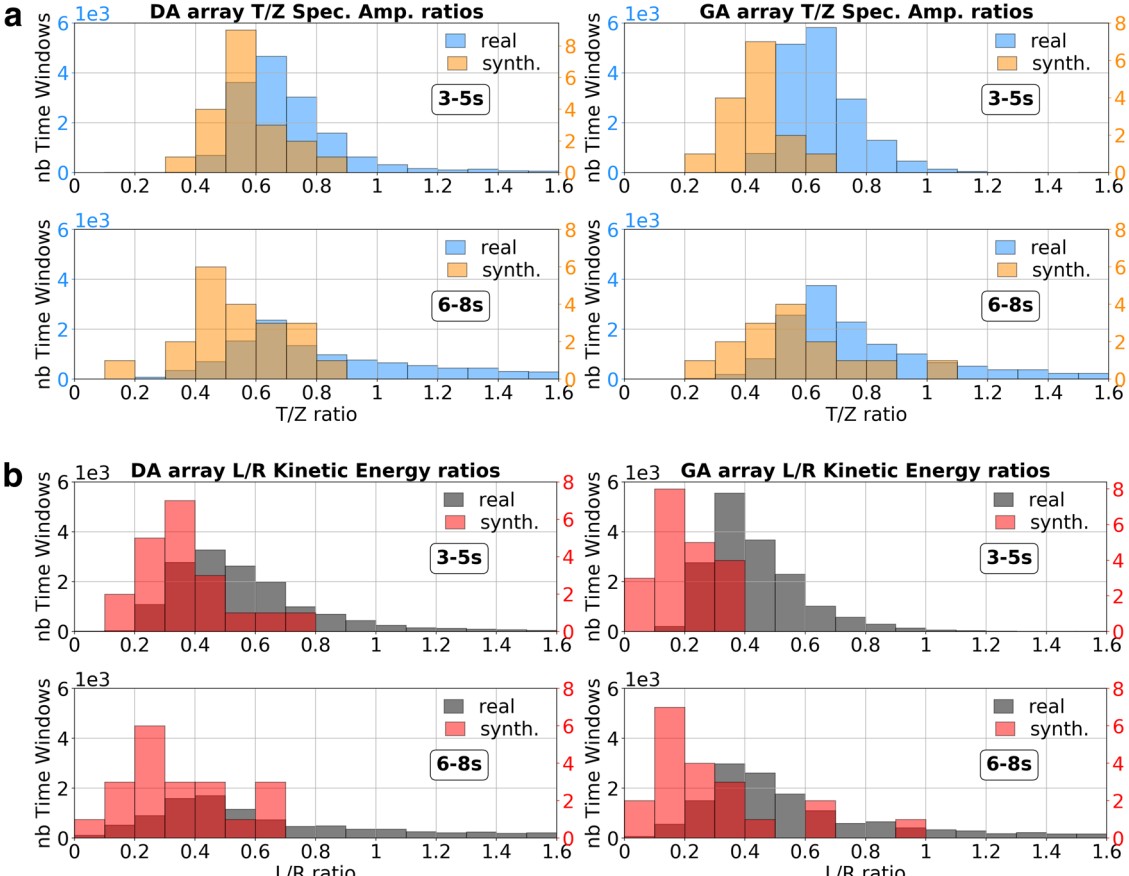

**Fig. 9 Irish Love to Rayleigh wave (L/R) ratios.** Comparison of observed and synthetic L/R ratios estimated for both DA and GA arrays. Since the L/R kinetic energy ratios will depend on a seismic velocity model beneath the arrays, the synthetic spectral amplitude ratios between vertical and transverse components are also shown. **a** Spectral amplitude ratios: the histograms show the distribution of the transverse/vertical spectral amplitude ratio for each time window considered in the array analysis. **b** Kinetic energy ratios: the histograms show the distribution of the L/R kinetic energy ratio for all time windows involved in the array analysis and are derived based on T/Z ratios and 1D eigenfunctions associated with a 1D velocity model beneath each array (Supplementary Fig. 19). For consistency, the same 1D models are considered for synthetic and real ratios estimations, but differ slightly for each array. More details on the ratios calculations are described in the "Methods" section.

diffuse back azimuth estimates for the wavefield observed at these longer spectral periods on the real data likely reflect a combination of the broader source distribution, and scattering effects from the sediments and deeper structures down to the upper mantle as the crust becomes very thin in some parts of the whole NE Atlantic hyperextended margin[30], not fully taken into account here due to the limited area covered by the 3D model.

The L/R ratios estimated from 3D simulation results are slightly lower than for the real data estimates over a year (Fig. 9), but both synthetic and real ratio distributions overlap well. In addition, similar to previous studies in Europe[10] and Australia[12], the ratios appear relatively constant over the year, particularly for the period band 3–5 s (Fig. 9 and Supplementary Fig. 24). This feature directly relates to the recurrent dominant back azimuths observed through the course of the year (Supplementary Figs. 1–3). Furthermore, the observed L/R energy ratios at the DA and GA arrays align quite well with the value of 0.5 previously observed for Ireland[10]. With most sedimentary layers located in deep water, it shows how the balance between a weaker source on the shelf and a stronger source over the deeper part of the model contributes to the amount of Love waves with respect to Rayleigh wave energy, observed on land. There are still some discrepancies that are likely related to the source approximation and discretisation, and also to the fact our 3D synthetic velocity model is also an approximation in both geological structure and

spatial area. A more heterogeneous model taking into account internal scattering[49] would also likely improve the L/R ratio comparisons. However, the simulation results show that the proposed mechanisms for Love wave generation reflect a major contribution to the L/R ratios observed in Ireland, on the N.E. Atlantic seaboard.

## Discussion
Multiple parameters will affect the relative levels of Love and Rayleigh wave energy observed on land: sediments thickness, velocity structure, basin morphology and lateral extent, bathymetry, station position relative to steepest slope orientations, and source distribution originating from ocean waves interactions. In that regard, each particular region around the world is unique and a detailed characterization of the environment is needed to understand the observed Love wave levels. However, interesting comparisons can still be made with our synthetic analysis (Fig. 6) to explain the observed L/R ratios from selected regions (Fig. 10). With dominant back azimuths pointing toward the North and South for sedimentary basins located along the coast of Australia, the role of sediments on the attenuation of Rayleigh waves and generation of Love waves has also been proposed to explain the observed ratios between Love and Rayleigh waves[12]. In Japan, the L/R energy ratios have been linked to sources along the

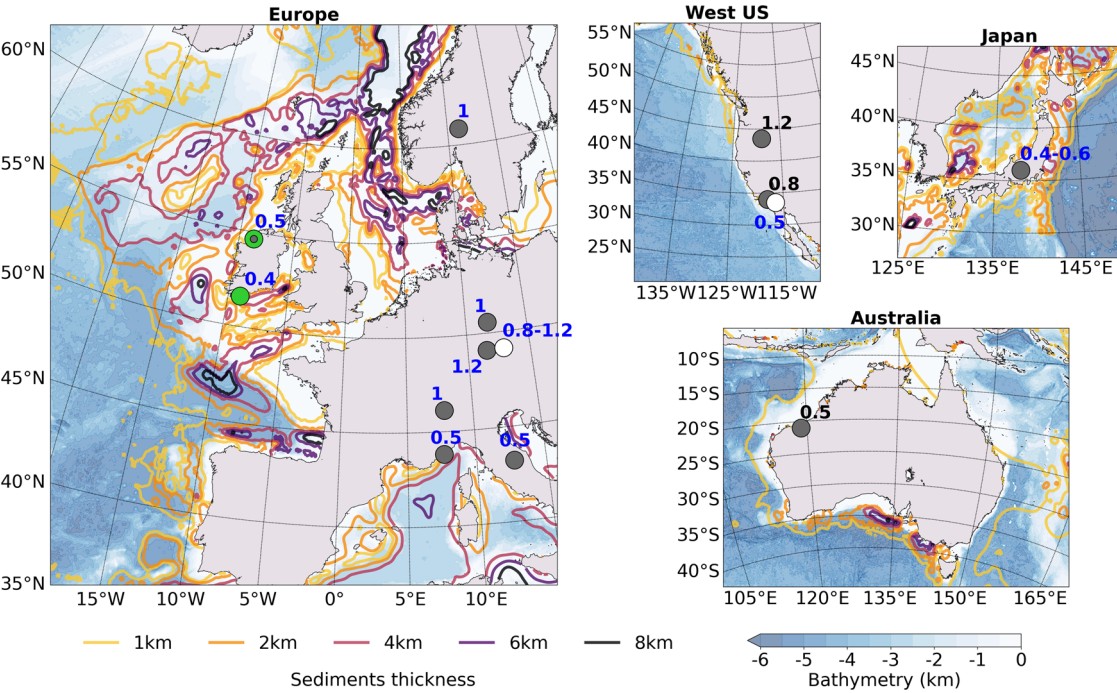

**Fig. 10 Love to Rayleigh wave (L/R) ratios around the world.** The ratios are compared with bathymetry and sediments thickness distributions[69] for several studies[9,10,12,15–17]: blue values are estimates based on Love over Rayleigh wave kinetic energy ratios and values in black are estimates based on beampower ratios. Europe: green dot ratios are from this study (3–8 s estimated over a full year)—gray dot ratios[10] have been estimated for dominant back azimuths over a full year for periods 6–8 s—white dot ratios[16] have been estimated over 5 years for periods 3–8 s. West US: gray dot ratios[9] have been estimated from R/Z beampower ratio for three selected days at 8 s—white dot ratios[17] have been estimated over four seasons for periods 3–10 s. Japan: ratios[15] estimated over half a year for periods 4–10 s. Australia: ratio[12] estimated from T/Z beampower ratio over a full year for period of 3 s.

continental shelf with dominants NE and SW back azimuths, depending on the seasons[15]. Those observations likely reflect 3D propagation effects from the very pronounced bathymetry gradient at the shelf, but also the sediments distribution around Japan with thicker sedimentary deposits in the NE and SW regions. For the West United States (US) region, L/R energy ratios comparable to Ireland have been observed[17], although larger ratios have also been derived for individual days linked with source interaction with the sediments located along the coast in the North[9] (Fig. 10). Europe represents an interesting case as it shows significant differences between L/R energy ratios averaged over long periods of times[10,16]: low ratios are found closer to the dominant NE Atlantic microseisms sources, whereas doubled ratios are observed in central Europe (Fig. 10). In fact, those observations agree quite well with the differences in ratios observed in our synthetic cases, showing high ratios and transverse amplitudes for thick, slow sediments located in shallow water (Fig. 6). It is worth noting that some of the thickest sediments distributions in Europe are found along the Norwegian coast and in the North Sea in relatively shallow waters. In addition, several studies have defined the shelf area along Norway as a substantial contribution in the secondary microseisms recorded in Europe[10,18,40]. With the NS orientation of the North Sea grabens, and associated low velocity sedimentary layers[50], a relatively high amount of Love waves generated in the North Sea compared to Rayleigh waves likely radiate straight toward central Europe, with no significant interaction from bathymetry changes, leading to the observed higher L/R estimates than for Ireland. In fact, for central Europe the dominant back azimuth directions linked with high L/R ratios point toward the NW[10], toward the North Sea.

We have only looked at 3D propagation effects in the ocean, but similar effects on land between the coast and a station in a middle of a continent will likely also affect L/R energy values, and therefore interpretations of microseisms processes. We have discussed the differences in the L/R values between Ireland and Continental Europe in terms of changing ocean source locations and geological environments beneath the seafloor, but we suggest that additional scattering along the inland path[49] can also enhance the "initial source values" that we see closer to the ocean. While Rayleigh wave excitation in deep water is largely controlled directly by ocean wave–wave interaction, we have determined that strong interactions occur at the shelf edge that leads to mislocations on beamformed arrays. Here, we see the origin of Love waves is determined by acoustic and seismic wave conversions along steep seafloor morphology, and in particular subsurface interfaces exhibiting high velocity contrasts, which are closely linked to the presence of sedimentary basins. In fact, the dominant generation mechanism appears to be associated with SH-to-Love wave scattering at sedimentary basin's edges right below the ocean source. In contrast, Rayleigh to Love waves conversions occurring at the seafloor along the microseism's path offer a more limited contribution. However, although we observe a structural control over the amount of Love waves generated in the ocean, Rayleigh wave amplitudes are more strongly modulated by the bathymetry and geological environment. Therefore, Rayleigh wave energy has a major influence in the Love to Rayleigh wave ratios observed on land. All these factors, lead to spatial and temporal sensitivities in Love wave observations that do not necessarily correlate with ocean wave states. However consequently they do yield spatiotemporal information about ocean–land coupling in both deep and shallow waters.

## Methods

**Array analysis**. The two Irish arrays consist of broadband (50 Hz to 60 s) Guralp 3-ESPDC seismometers (sampling rate 100 Hz); with DA having an aperture of 26

km and a mean interstation spacing of 12 km; and GA having an aperture of 29 km and a mean interstation spacing of 15 km. Data availability at the arrays for the analysis time period is described on Supplementary Fig. 25. The geometry and theoretical array response for both arrays are detailed in Supplementary Fig. 26. For the array analysis, we use Capon's high-resolution frequency wavenumber (HRFK) method[51], which is implemented in the Geopsy software package[52–54]. The Geopsy software is freely available and can be accessed at http://www.geopsy.org/download. php. For each propagation direction, the horizontal component signals are rotated into radial and transverse directions, and the HRFK processing is then performed on vertical, radial, and transverse components[55]. The f–k processing consists of delaying the observed recordings at different stations according to a particular horizontal wavenumber vector, $\mathbf{k} = (k_x, k_y)$, and computing the beampower as follows,

$$BP(\omega, \mathbf{k}) = \frac{1}{\mathbf{e}(\omega, \mathbf{k})\underline{R}^{-1}(\omega)\mathbf{e}(\omega, \mathbf{k})} \qquad (1)$$

where $\underline{R}^{-1}$ is the inverse of the spectral covariance matrix and $\mathbf{e}$ is the steering vector. The symbol represents the conjugate transpose operation. Since the inverse of $\underline{R}$ is used, it is necessary to ensure the matrix is non-singular. The diagonal loading of the covariance matrix is applied to ensure the eigenvalues of $\underline{R}$ are bounded by the diagonal loading value[51,56]. For consistency, we keep the same algorithm for all array analysis throughout the study, unless specified otherwise. More details on each analysis case are detailed below.

- *Real data array analysis* (Fig. 1): the data from each array were corrected for instrument response and downsampled to 1 Hz. We use a sliding time window of 3600 s with 50% overlap and calculate f–k spectra for two frequency bins between 0.2–0.33 and 0.12–0.2 Hz, with respective central frequencies at 0.27 and 0.16 Hz. A diagonal loading parameter of 0.001 was used. From the f–k spectrum, back azimuth, slowness, beampower, and a unitless semblance coefficient[57] are calculated from the maximum of the f–k spectra for each time step and frequency bin. For the triangulation of surface wave source locations, slowness values were limited to the range 0.25–0.33 s/km for both arrays. The semblance threshold used for plotting the source distribution maps is 0.15 for the DA and 0.2 for the GA.
- *Synthetic data array analysis—"extended source"* (Fig. 8): for consistency, we used exactly the same approach as for the real data, except here we used a sliding time window of 200 s with 50% overlap. The 200 s window length was chosen as a compromise between considering a relatively long noise signal, but relatively short to be able to build the source distribution maps based on a few back azimuths estimates and to create a distribution of L/R ratios from the analysis.
- *Synthetic data array analysis—single point source* (Fig. 3): here, since we know the source signal is transient, we used the conventional f–k approach from Geopsy with a smaller time window of 10 s with 50% overlap. The better temporal resolution allows the identification of particular seismic phases along the signal trace recorded at each array.

**Source distribution maps derived from array analysis.** In order to obtain source distribution maps from the HRFK data, we aim to first develop distributions of the back azimuth measurements at each array; then project these distributions onto a map; and finally combine the distribution maps from the two arrays. For the back azimuth distributions two important considerations must also be accounted for:

(i) The microseism wavefield observed at an array is complex and can contain multiple coincidentally arriving signals. This implies the source distribution may be multimodal, hence a nonparametric method is required to estimate the distribution.

(ii) The back azimuth is an angular quantity meaning the distribution needs to be defined on the unit circle.

The first point is dealt with by constructing the back azimuth distribution using kernel density estimation. The kernel density estimate (KDE) is a commonly used nonparametric method to determine the distribution of a data set. The distribution of data points (here the back azimuth) can be illustrated through a single variable plot, such as a rug plot (Supplementary Fig. 27a), which plots the back azimuth as vertical lines along the x-axis. To develop a KDE, we wish to fit a kernel function to each data point. As back azimuths are a circular quantity we require a kernel that is defined on the unit circle. The von Mises distribution is a close approximation to a wrapped Gaussian distribution and is an appropriate choice for the kernel (Supplementary Fig. 27b). The probability density of the von Mises distribution is defined as:

$$p(x) = \frac{e^{\kappa\cos(x-\mu)}}{2\pi I_0(\kappa)} \qquad (2)$$

where $\mu$ is the mode and $\kappa$ the concentration (the reciprocal of dispersion $1/\kappa$ which is comparable to the variance), and $I_0(\kappa)$ is the modified Bessel function of order 0. The kernel functions are then summed and the resulting distribution is normalized so that the maximum has a value of one creating a relative distribution. If the chosen kernel is smooth then the resulting distribution is a smooth integrable

function (Supplementary Fig. 27c). These relative distributions are calculated in 3 h windows for each array.

We then project these distributions onto a 0.1 by 0.1-degree spatial grid (Supplementary Fig. 28a). For each point on the grid, the back azimuth along the great circle path to a reference point (the array center coordinate) is calculated. This theoretical back azimuth is then used to assign every grid node a density value based on the distribution obtained from the KDEs (Supplementary Fig. 27). The projected distributions are then combined into a single map by taking the product of the DA and GA projections (Supplementary Fig. 28b). This combination ensures that only sources observed by the two arrays are included in the final map. The resulting source density distribution map is normalized so that the maximum has a value of one creating a relative distribution. This approach is used on all components and two-period bands (3–5 and 5–8 s). The source distribution maps of Figs. 1 and 8 are generated using the back azimuth data of Supplementary Figs. 1–3 and synthetic back azimuth data of Supplementary Figs. 21–23, respectively.

**3D Numerical simulations.** The SPECFEM3D[31–33] Cartesian software is a freely available code and can be accessed at https://geodynamics.org/cig/software/ specfem3d/. For the SPECFEM3D simulations, a polynomial degree of $N = 4$ is used to represent functions in each spectral element, and therefore each element contains $(N + 1)^3 = 125$ Gauss-Lobatto-Legendre (GLL) integration points. At the edge of the models, the Stacey absorbing conditions were used in the simulations. They are not perfect but for this study and the size of the models used, the side reflections are considered to be effectively negligible.

- *3D concept model*: The model is defined on average by ~1.8 million elements depending on the sediment, bathymetry structure and extends to 60 km depth. Three different units that are included are water, sediments, and crust. A minimum resolution of 3.3 s is obtained for simulations, including very slow sediments ($V_p = 2$ km/s). The minimum period resolution is defined by the minimum wavelength that the 3D mesh can resolve in the simulation without introduction of numerical dispersion. It is therefore mainly dependent on the velocity model. The simulations are performed over 200,000 time steps with a time step value of $dt = 0.004$ s (sampling rate 250 Hz). The simulations were run on the Irish Center for High Performance Computing (ICHEC) KAY cluster, as well as DIAS computation resources. They did not include anelastic attenuation, except when specified.
- *3D regional model of the Irish offshore*: the model is composed of ~4.9 million elements with a minimum period resolution of 3.4 s, thus resolving most of the whole secondary microseism spectrum (3–10 s) within the mesh and associated velocity model. Four different units are included: water (acoustic medium), sediments (viscoelastic medium), crust (viscoelastic medium), and mantle (viscoelastic medium). The bathymetry is derived from ETOPO1 Global Relief Model[58]. The sediment basement depth is taken from NOAA sediment thickness world map[59]. The Moho depth was taken from the EPcrust[60] model. The projection considered for the model is UTM zone 28 N. Although only a maximum depth of 40 km is shown in Fig. 2, the 3D model used in the simulation extends to 60 km. The 3D velocity model (Fig. 2) is based on $V_p$ velocity values from several studies covering the NE Atlantic[29,30,61,62]. Empirical relations were used to extract $V_s$ and density values[63], as well as attenuation $Q_p$ and $Q_s$ (Supplementary Table 1) assumed constant over the frequencies of interest[64]. The point source simulation (Fig. 3) is performed over 320,000 time steps, whereas the "extended source" simulation (Fig. 8) is performed over 680,000 time steps both with a time step value of $dt = 0.003125$ s (sampling rate 320 Hz). For the simulations, the whole mesh is decomposed into 320 mesh slices for MPI parallel computing. The two simulations performed for the regional model include anelastic attenuation.
- *P2L source*: the source used in the "extended source" simulation (Fig. 8) includes multiple point sources derived from the equivalent surface pressure or microseism source excitation (P2L) data resulting from nonlinear wave–wave interactions[24]. The P2L data are available through the Ifremer (French Research Institute for Exploitation of the Sea) ocean wave numerical model built within the WAVEWATCH III (ref. [65]; WW3) modeling tool, as part of the IOWAGA project[66]. They characterize the microseism excitation as a function of latitude, longitude, frequency, and time (every 3 h). The data can be obtained through ftp://ftp.ifremer.fr in the directories: /ifremer/ww3/HINDCAST/GLOBAL/ 2016_ECMWF/p2l for the year 2016 data and /ifremer/ww3/HINDCAST/ GLOBAL/2017_ECMWF/p2l for the year 2017 data.

The goal here is mainly to simulate a realistic secondary microseism source spectrum distribution over the study area, so we do not attempt to perfectly match the absolute source amplitude with real world amplitudes. In the "extended source", the acoustic source amplitude is given by the median of the P2L spectrum over a year (March 2016–March 2017). The median P2L data are discretized into source points with 12.5 km spacing for an area covering most of the model. At each grid point, the P2L spectrum is resampled in order to extract pressure time series with a sampling rate high enough to match the sampling rate required by the simulation. The time step used for the simulation is $dt = 0.003125$ s. It is constrained by the size of the mesh elements in comparison with the wavelength of the simulated signal, in order to have a numerically stable

simulation. Here, the external source defined by the pressure time series requires the same sampling rate. Since, the downloaded P2L data in the frequency domain have a coarse sampling rate (orange dots in Fig. 8a), it needs to be resampled in order to have a P2L spectrum with frequency resolution d$f$ = 1/($dt \times N$), where $N$ is the number of time steps in the simulation ($N$ = 680,000). After interpolating the P2L frequency spectrum at each geographical grid point and introducing a random phase, the P2L spectrum can be transformed to a pressure signal for each grid point. This leads to a total of 3097 points sources (Fig. 8a) continuously injecting acoustic noise signal in the simulation for a total recording time of 2125 s ($N \times$ d$t$).

**Love to Rayleigh wave energy ratios**. The Love to Rayleigh wave energy ratio estimations follow the same approach as previous studies[10,15–17,49] by determining the ratio between Love ($E_L$) and Rayleigh ($E_R$) kinetic energies:

$$\frac{E_L}{E_R} = \int_0^{z'} \rho W(z)^2 \mathrm{d}z \left( \int_0^{z'} \rho (\mathrm{U}(z)^2 + V(z)^2) dz \right)^{-1} \qquad (3)$$

In the equation, the terms $W$, $U$, and $V$ represent the respective Love, Rayleigh vertical, and Rayleigh radial eigenfunctions, $\rho$ being density and $z$-depth. The Love wave ($W$) and Rayleigh wave vertical ($U$) components are scaled based on the spectral amplitude ratio T/Z between transverse (T) and vertical (Z) components for the data of interest. An example of T/Z calculation is shown on Supplementary Fig. 18. The obtained T/Z spectral amplitude ratios averaged over two-period bands 3–5 and 6–8 s are then used to scale the Love and Rayleigh fundamental mode eigenfunctions derived from a 1D velocity model below the stations of interest (Supplementary Fig. 19). The eigenfunctions were determined using the "computer programs in seismology" software package[67] and particularly scripts sregn96 and slegn96. For L/R kinetic energy ratio estimates based on simulations from the 3D concept model, we consider the full 800 s signal recorded for the calculations of the T/Z ratios.

For the L/R estimates from synthetic and real array analysis, we consider each time window defining the array analysis process, separately. This means the L/R estimates are calculated for windows of 3600 s for the real data and 200 s for the synthetic data. Vertical and transverse channels were beamformed based on the back azimuth and slowness associated with the maximum beampower for each time window. This enables to determine the distribution of T/Z ratios over period bands 3–5 and 6–8 s from all the beamformed signals at each time window. The T/Z ratios are then used to calculate the L/R kinetic energy ratios, as described above.

The toolbox "ObsPy"[68] was used for handling the data and associated seismological processing.

## Data availability
The data that support the findings of this study are available from the corresponding author upon reasonable request.

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

## Acknowledgements

This publication has emanated from research supported in part by a research grant from Science Foundation Ireland (SFI) under Grant Number 13/RC/2092, and co-funded under the European Regional Development Fund and by PIPCO RSG and its member companies. We acknowledge the Irish Center for High Performance Computing (ICHEC) for access to the KAY cluster for the numerical computations, as well as Fabrice Ardhuin and Ifremer for the availability of the P2L data through the server ftp://ftp. ifremer.fr. Finally, we also would like to thank the developers of the SPECFEM3D and Geopsy softwares.

## Author contributions

F.L.P. designed the models and performed the numerical simulations. D.C. completed the array analysis. C.J.B. initiated the concept and supervised the research. All authors wrote the manuscript.

## Competing interests

The authors declare no competing interests.
