## [Peer Review File · Nature Communications]

REVIEWER COMMENTS

Reviewer #1 (Remarks to the Author):

The manuscript addresses the difficult question of the generation of Love waves in the seismic noise wavefield. It first illustrates how beamforming back projects the Rayleigh noise sources to the shelf rather than to the deep sea, as observed in data from two arrays in Ireland. Second, using 3D numerical simulations in a realistic regional crustal model (including water layer and sediments), it convincingly demonstrates that this location of surface wave generation can be explained by propagation effects related to the shelf edge.

The MS then further explores the mechanisms behind the Love waves through numerical simulations in simple models. This section has great potential, but could be better explored and partly lacks clarity (see below).

Finally, the MS models the wavefield for a realistic distributed source distribution and compares observed Love/Rayleigh amplitudes and energy with simulated ones.

--

The study has a number of strong points:

- It brings interesting new insight into a complex problem for which the literature is sparse.
- It is fundamentally sound, with no demonstrable errors (although it may not fully exploit its potential).
- The 3D simulations are interesting; I have not seen anything of similar quality for continental margins.
- The subject is in itself of general interest: A better understanding of their origin gives insight into wave propagation, coupling of different wave types at the sea bottom, and can be beneficial for better imaging using Love waves present in the seismic noise.

This work should be published, whether in Nature Communications (see point 1 below) or elsewhere. My suggestions should be seen in the light of further promoting the impact of this work, which I consider of high quality. I therefore recommend major revision, but point 1 below may need special editor attention.

I have two main comments:

1. The authors show that there is a big difference between the observed Love wave amplitudes and the simulated ones. They invoke mechanisms of why this is the case, but none of these additional contributions are quantified. Also, there is an additional problem because the fundamental mode Rayleigh waves tend to be mainly forwards scattered, with relatively small coupling to Love waves or higher modes (see for example work by Snieder, 1986, 1988). Due to the mismatch between observations and simulations, the manuscript does therefore not carry a full answer to the enigma of the strong Love waves. This was also the case for the 'continental' study by Pedersen et al., cited by the authors. The MS rather confirms and quantifies a mechanism in wave propagation that seismologists know in general terms, and a mismatch with observations which specialists in surface wave studies have been frustrated for some time, previously in the context of 'tectonic release' of nuclear explosions, but now also in the area of seismic noise studies.

To make the study of more general rather than specialist interest, other areas for which Love wave source locations have been identified could be included (but this would be a huge work) to check whether the conceptual model presented by the authors works elsewhere. The authors could also further quantify the additional effects that are presently just mentioned as a potential explanation to the mismatch between observations and simulations.

If the hint of Love wave generation being efficient only when the source is located above a strong heterogeneity (see below), it should be possible to relate changes in transverse energy with high waves in very specific areas. This would probably be a frequency dependent effect. If that type of geographical effect was demonstrated, that would bring a very interesting insight into Love wave generation for seismic noise.

While in the present stage I am not certain the manuscript is sufficiently general to warrant publication in Nature Communications, I would recommend to give the authors a chance to further improve and generalize the study.

2. My second comment is related to the simple models, presented in Figures 4 and 5 and associated text. As I understand it, their scope is to understand the mechanisms of the Love wave generation, but this section needs quite some improvement to fully enlighten the reader.

As I understand the text, the idea of the simple cases is to achieve an in-depth understanding of the wavefield. I have spent a few hours trying to relate specific arrivals in the seismograms of the figures with specific propagation paths and/or wave conversions, but so far I was only partly successful. Some of the comments and questions could potentially be addressed through snapshots of the wavefield in supp. Mat. or for example through more clear explanations in the text, identification of different wavetrains etc.

Below I list some of the immediate questions and suggestions that spring to mind, but there are surely other key information and conclusions that could result from more in-depth analysis of the 'simple' cases.

a. For me, one of the most interesting observations is that the Love waves seem to be present even in a very simple geometry (bathymetry part of fig 4), if the source is located in a laterally heterogeneous area. This is clearly illustrated by the case S1-R1. This idea may possibly be a key ingredient for the Love wave amplitudes. Indeed, looking at the case S2-R1, the Love wave seems to be minimal (waves around $t=40s$, if I got distances and velocity right). This would mean that the case S1 is more favourable than S2. Is there any way to explore this idea further?

b. Again, if I got the times right: what is the large second wave train? Does it correspond to the slowly propagating wave on the left side of S1 in the snapshot? If so: could the high amplitudes on the transverse component for that wavetrain be due to a simple change of propagation direction due to change in wave speed? (especially as it seems to arrive at the same time as the waves on the vertical and radial components). Which arguments do the authors have to say that the late arrival on the transverse component is really a Love wave? I don't think the timing would be explained by different modes, for example, but if it is the case, the MS needs to have a clearer and quantified explanation, with information on higher mode wave velocities at the dominant frequency.

c. The same observation seems to hold for the Sediment case, where the S1-R1 case has a clear and early arrival on the transverse component (even though the vertical and radial scales are different from that of the transverse component – I would btw suggest to have identical vertical scales on all components, considering the question at hand).

d. For this case, the S2-R1 case seems to have early energy on the transverse component (attention vertical scale) for the late arrivals, but hardly for the early ones. What is the interpretation of this?

e. The case of S3 is interesting: the source is in a very heterogeneous area. The Rayleigh waves have a very long duration, but there is significant energy on the transverse component. However, in terms of energy, the increase of duration of the Rayleigh wave means that the total energy on the transverse component is very small as compared to the vertical and radial components. Are there other frequency intervals where the transverse component demonstrates more resonant behaviour in

the sedimentary basin? Could such resonance effects be of interest with Love wave generation? What if the lateral heterogeneity coincides with a depth of resonance for the Love waves?

f. In figure 5, it is necessary to explain which wavetrains were used for the analysis, because of the issues with the different wavetrains discussed above.

g. Also in Figure 5: Are the variations between azimuths and models more related to increase/decrease in R&Z amplitude or increase/decrease in T amplitude, or a combination of both? Between models 1 and 2: are the differences related to change of resonance frequency of Love and Rayleigh waves in the basin?

Minor comments

3. Distances could be shown in km rather than m. It would be nice to know the precise coordinates of sources and receivers; it is quite difficult to figure out based on the figures (or did I miss something ?)

4. We should be informed of the propagation velocities of Rayleigh and Love fundamental mode waves in each part of the two models, for the dominant frequency (table would be fine). This would make it more easy to relate wave arrivals with specific waves.

5. I would suggest to change numbering between R1 and R2. This means that the numbering would correspond to a progression in discussion from the simplest case (presently S1-R2) towards increasingly complex cases.

6. For figure 5, I would suggest that the authors delete model 3. It is a useful test, but can be mentioned in the text only, and it complicates the figure unnecessarily.

7. I really appreciate the layout and content of Figure 7

8. Only the hours where the backazimuth is defined within $\pm 10^\circ$ of the dominant beam direction over the full year were considered. It is not easy to understand the impact of this choice: so we are looking at Love waves that are the closest of the situation in the simulations? A clarifying sentence would be helpful.

9. Supp fig 4 does not need to have the vertical axis go as far as 3. Two would be sufficient.

10. Supp fig 6 : please give definition of propagation direction (I assume it is not clockwise angle from north of the incoming wave).

11. Double check formulation and text for all figure captions, there are a couple of misses and bad sentences.

12. Some figure text is too small

13. Potential references to add or have a look at:

- Approximation of surface wave mode conversion at a passive Continental margin by a mode-matching technique, T.Meier and P.G. Malischewsky, GJI 2000 : there are so few papers on the simulation of Rayleigh/Love waves at continental margins that the authors should include this one.
- S-to-Rayleigh wave scattering from the continental margin observed at USArray, Buehler J.S, Mancinelli N.J, Shearer PM, GRL 2018 could be of some interest?
- Imaging lateral heterogeneity in the northern Apennines from time-reversal of reflected surface waves, Stich et al., GJI 2009 : also has additional insight even though not from continental margin.
- Influence of the seismic noise characteristics on noise correlations in the Baltic shield, Pedersen et al., GJI 2007: supports the results of the present study, as it shows that the 2nd microseismic peak Rayleigh waves may be distributed along the continental margin, as compared to a more focussed noise direction in the first microseismic peak.

Reviewer #2 (Remarks to the Author):

How deep ocean-land coupling controls the generation of ocean microseism Love waves

Florian Le Pape, David Craig, Christopher J. Bean

Le Pape and co-authors report on a study of the generation of Love waves in the secondary microseism. Although many observations have shown the presence of Love waves in the microseismic wave field, no clear generation mechanism has been found so far.

Here, the authors use numerical simulations show that conversions from Rayleigh waves can result in microseismic Love waves, in the case of some specific bathymetry and subsurface configurations inspired by the regions offshore of Ireland.

By simulating a few examples, the authors show that factors such as bathymetry, sediments and the geometry thereof has an effect on the amount of Love waves generated. The analysis is not very systematic and the interpretation stays mostly qualitative, but it is enough to show that the composition of the transverse polarized surface wave field can be a complex interplay of many factors.

This confirms what the literature about the secondary microseism has been suggesting so far, and it is one of the few studies that uses forward modeling in their investigation. In that sense, the results are of interest to the scientific community. The manuscript contributes to our understanding of how the composition of the secondary microseismic wave field emerges.

The manuscript presents a lot of interesting results (such as, for example, the strong azimuthal dependence of L/R ratios, the complexity of the wave field in the presence of sediments, etc), which unfortunately are not discussed in depth. This is likely due to constraints on the length of the article. In my opinion, the paper would also benefit from a bit more discussion of the relative contribution of different factors (sediments, bathymetry, source) to the excitation of Love waves.

The paper is well written and the figures present the results well enough. A few more figures could be included in the supplementary material to highlight some interesting phenomena.

Two main things raise some questions about this manuscript:

__First__, the assumptions about the source locations, and subsequent comparison of simulation results, are based on observations of the secondary microseism in Ireland.

Not enough information is given in this manuscript to render these observations reproducible. E.g. no information is given about the arrays used (geometry, instrumentation, ..). Beside a general description of the beamforming method, not much is detailed about the processing of the three-component array data.

So based on the information given in this manuscript, it is not possible to evaluate the reliability of these observations, which makes it difficult to evaluate how well the simulations presented here are able to reproduce real-world observations.

I suspect that the observational study is presented in a different manuscript which appears to be still under review. This 'companion' paper [Craig & Bean, G3] is cited a few times, with statements that seem to support the results presented here:

I.80: "Those observations cannot be explained by ocean wave models only, without taking into account seismic wave propagation effects [23].

I. 144: "The simulation results are very consistent with real array findings where body waves can be seen in deep water[23] and surface waves locate at or near the shelf break (Fig. 1)"

One wonders why the authors did not merge the two publications. It seems that the results presented in this manuscript depend strongly on the results presented in [Craig & Bean].

[23] Craig, D. & Bean, C. J. Double Frequency Microseism Sources Observed with Seismic Arrays and an OBS Network in Ireland. *Geochem. Geophys. Geosyst.* __in review__, (2019).

__Second__, in figure 4, the arrival times for some of the transverse components are confusing. In 4c, for the S1-R1 path, the transverse component arrives before the radial and vertical component arrivals. However, for S2-R2, all three arrivals seem to share the same traveltime. I would expect the transverse arrival to come visibly earlier than the other two, in the case of a true Love wave generated either at S2 or converted at the slope (at a distance of >50km). In figure 4e, it can similarly be noted that all three components for the S2-R1 path share the same traveltime.

This raises the question whether these arrivals are 'true' Love waves, or perhaps something like a quasi-Rayleigh wave? Perhaps a Rayleigh wave from a different azimuth than assumed when rotating the signals? I could imagine such perturbations of the polarization happening in the presence of a sedimentary basin or a steep slope.

This could be investigated by considering the wave field snapshots or animations. Alternatively, using arrays of receivers in the simulations would allow to distinguish the surface wave polarization and the corresponding slowness. The authors could either use the f-k analysis presented in the methods section, or an approach such as presented in Gal et al., 2018 (cited).

Comments on main text and figures:

I. 31: "The occurrence of Love waves in recorded primary microseism signals has been explained by the interaction between propagating ocean wave and sea-bottom topography"

Also consider the following two papers:

Fukao, Y., Nishida, K., & Kobayashi, N. (2010). Seafloor topography, ocean infragravity waves, and background Love and Rayleigh waves. *Journal of Geophysical Research: Solid Earth*, 115(B4).

Ardhuin, F. (2018). Large-scale forces under surface gravity waves at a wavy bottom: A mechanism for the generation of primary microseisms. *Geophysical Research Letters*, 45(16), 8173-8181.

I. 45: "with observed Rayleigh to Love wave energy ratios ranging from 0.4 to 1.2 or as low as 0.25."

Additional ratios are presented here:

Tanimoto, T., Lin, C. J., Hadziioannou, C., Igel, H., & Vernon, F. (2016). Estimate of Rayleigh-to-Love wave ratio in the secondary microseism by a small array at Piñon Flat Observatory, California. *Geophysical Research Letters*, 43(21), 11-173.

I. 54: "However, the source locations derived from ocean wave models often do not agree with actual source locations determined from land-based seismic observations." References?

Figure 1: What is the configuration of DA and GA arrays? How is the probability density obtained from the array analysis? What is the influence of the array response on the extent of the probability density

maximum? How variable is the source location over the course of the year?

I. 73-75: "One significant advantage of these arrays is their location along the Irish coast on the North East (NE) Atlantic seaboard. There is no "contamination" along the terrestrial seismic wave propagation path from inland structures"

Do you expect any effect of the cliffs along the Irish coast? As suggested by Friedrich et al. 1998 (cited), steep coastal structures could contribute to the generation of Love waves.

I. 115-118: "We start by considering a single point source located 15m below the sea surface and defined by a Gaussian wavelet source time function of duration 3s, leading to a relatively flat spectrum over the secondary microseism periods"

The source mechanism employed in the first simulations seems to be an acoustic point source (explosion?) at the top of the water layer. Generally, for the secondary microseism, a vertical pressure source (as proposed by Longuet-Higgins, 1950, cited) is assumed. I suggest that the authors argue why their choice of source mechanism is reasonable.

Figure 3: please specify what the green star represents.

I.181: "Both Figure 4d and 4c show that the radiation pattern" is figure 4b meant?

I.183: "The transverse wave field radiation patterns are significantly affected by the source location, the lateral geometry of the shelf edge and changes in its slope steepness (Supp. Fig. 2)."

I disagree: from Supp. Fig. 2, the two values of slope steepness tested barely affect the radiation pattern, and only changes the amplitude of the resulting transverse wave field.

I.193: "Similar to the "bathymetry-only" model (Fig. 4a), the shape of the sediment edge will also significantly affect the radiation of the transverse wave field and therefore the Love wave energy detected by two different receivers."

This is not shown anywhere. Such a statement should be backed up with simulation results, even if only in the supplementary materials.

I.201: "Heterogeneity is introduced by randomly adding ± 0.2 km/s on the velocity values of model 1."

Is this value based on real velocity models? What is the scale of the heterogeneity (especially compared to the wavelength)? Is the heterogeneity self-similar, as is the case for many earth materials?

figure 5: The results presented in figure 5 are interesting, yet are hardly discussed. As cited in the introduction, L/R ratios are known from observations in other studies. The authors could at least comment on how their values compare.

Moreover, Gal et al. 2017 (cited) extensively considers the influence of sediments on the secondary microseism wave field. The authors could discuss how their results relate.

I.231: "used as a time synchronised input source grid for the 3D numerical simulation": I assume you are using the same point sources as before at each grid point?

I. 278: "as well as a more heterogeneous models" singular or plural?

figure 6a: the way the different periods are presented, the "strongest ocean source area" in the NW (I.237/238) is hidden, unfortunately.

I.304-307: "We demonstrate that multiple parameters will affect those Love wave levels... and source distribution originating from ocean waves interactions."

Where is this shown?

I.312: "implying that scattering along the path can enhance the "initial source values" that we see closer to the ocean."

This is supported by Ziane, D. & Hadziioannou, C. 2019 (cited)

Comments on the Methods section:

Not enough detail is provided for the study to be reproduced, in particular the signal processing is not fully explained. How is the back-azimuth, slowness etc obtained separately for radial and transverse components in the array analysis?

How are the probability density maps obtained?

I.382-383 please define the symbols used in the equation.

Comments on the supplementary material:

figure 4 & 5: since the radiation pattern (azimuthal distribution of ratios in figure 5) and signal structure (waveforms figure 4e) seem to be so complex for the model with sediments, I think it would be very interesting to show a snapshot or animation of the wave field, preferably from source S3. This could be included in the supplementary material.

supplementary figure 5: please specify what the different colored lines (green, red) represent.

supplementary figure 6: what is meant with propagation direction? Backazimuth, or backazimuth + 180 degrees?

Comment on figures, general:

To keep your figures accessible to colorblind users, please consider using appropriate colormaps (e.g. no rainbow colormap) and differentiating lines with line styles in addition to color.

Reviewer #3 (Remarks to the Author):

Review comments on Le Pape et al., submitted to Nature Communications

This paper shows possible excitation mechanisms for Love waves of secondary microseisms, which are mainly related to bathymetry and sediment structure in the ocean areas, using both large-scale three-dimensional numerical simulations of seismic wavefield and observed data in Ireland. The demonstration of Love wave generations from the numerical simulations and observations are reliable and interesting, and this paper has an important contribution to our understanding of excitation mechanisms for secondary microseisms. I recommend publication with some revisions, as discussed below.

Main comments

1) On multiple modes of surface waves in the synthetic calculations

First of all, it is better for readers to define the observed signals (Fig. 4). In particular, there are two signals in the right panel of Fig. 4c, while one pulse can only be seen in the left two panels of Fig. 4c. The authors stated that those two waves are the fundamental and higher mode (L180), but it seems

that more detailed descriptions are required, as shown in Fig. R1.

In addition, I do not understand wave propagations in the T component for S3-R1 and S3-R2 (Fig. 4e and Fig. R2), because their travel times are significantly different from the Rayleigh wave ones. Please explain more details on the propagations.

(Suggestion 1) I recommend that the authors add some figures showing record sections (waveforms aligned as a function of distance) at each component for S3-R1 and S3-R2 with putting more receivers, e.g., like light-blue and red circles in Fig. R2. It is expected that these plots show propagation velocities of the observed signals and also the conversions of surface waves at the slope (continental shelf), particularly for Love waves. If possible, such record sections for the case in Fig. 3 are also helpful for readers.

Secondly, one of the important findings of this study is that, in Fig. 4, the two Rayleigh waves have comparable amplitudes, which have not been observed in two-dimensional numerical simulations in previous studies. Can these features be seen at other propagation directions for Love waves in the simple models or the complex 3D model? If yes, when estimating L/R energy ratios, the amplitude balances of these two waves are very important. (I mean that, for some geometries between sources and receivers, the former amplitudes of the Rayleigh waves in the Z component are larger than the latter amplitudes, while the former ones of the Love waves in the T component are smaller than the latter ones, and vice versa. Such amplitude balancing may affect the estimation of L/R energy ratios.)

(Suggestion 2) *This is just based on my curiosity, so the authors do not need to reply this. When heterogeneities in model 3 (Fig. 5) have anisotropy in correlation lengths between the x and y axes, are the radiation patterns of Love waves and their higher modes different? I just suppose that, if there are some mechanisms that change the radiation patterns of the modes, it is very interesting and possibly affects L/R energy ratio estimations.

2) On seismic frequency ($2f$) and ocean wave frequency (f)

Secondary microseisms are excited by wave-wave interactions of ocean swell, but the secondary microseisms have double frequencies compared to those of ocean swells. The authors used P2L between 4 s and 10 s in Fig. 6, but which do those periods correspond, $2f$ or f ? It is nice if the authors explicitly mention this in the text.

Minor comments

L9-11

It is better to state that this sentence corresponds to secondary microseisms.

L18

Love waves do -> Love waves observed in land do

L81-82

In the period of 5-8 s, the source locations estimated using the Z and T components are different, so that I confused this sentence after checking Fig. 1.

Caption of Figure 2

-L98 What is "a min period resolution"? If the sampling rate of the synthetic waveforms is 0.33 Hz (3 sec), it seems that the authors can use up to 0.166 Hz, which is lower than the frequency band of secondary microseisms.

L119

located above a -> located in a

Caption of Fig. 3

-Please explain more details, including red regions, light-blue stars, and red triangles in Fig. 3a and red squares in Fig 3b.

-located just below the sea surface -> located 15 m below the sea surface

-Please remove "mechanism" in L131.

L140

Please define "elastic Rayleigh waves in shallow water". I think that, even 300 m water depth, the observed surface wave is the pseudo-Rayleigh wave.

L142

the wavefield apparent propagation direction -> apparent propagation direction of the wavefield

L181

Figure 4d -> Figures 4b

L237

I cannot understand the location of the strongest ocean source area in Fig. 6b. Please identify it with a symbol.

L249

It is better to specify the directory of P2L in the ftp site.

Fig. 6b and L258

The snapshot of the wavefield is unclear. Please improve it.

L301

I agree that a dipping interface at the bottom of sediments potentially produces Love waves within marine sediments in addition to Love waves whose energies are concentrated in the underlying crust. In numerical simulations, are such sedimentary Love waves transmitted to land? I am interested in reflection and transmission of Love waves at the sediment boundary slopes, and whether such transmitted Love waves affect L/R energy ratios. It is nice if the authors state on weak or strong transmissions of Love waves at the sediment boundary slope in the text.

L341

The authors only explained the method of the numerical simulations using the complex 3D structure, and did not explain the bathymetry and sediment models in details. In particular, it is better to explain the heterogeneous structure in model 3 (e.g., correlation length and other parameters, and why the authors chose them here) and the sediment structure in models 1-3. It seems that the sediment structure and the location of S3 represent the situation on the T component case of 5-8 s in Fig. 1.

L382

For the equation, please define the variables used.

Throughout the paper

Please unify from meters to kilometers in the text and figures

REVIEWER COMMENTS

Reviewer #1:

The manuscript addresses the difficult question of the generation of Love waves in the seismic noise wavefield. It first illustrates how beamforming back projects the Rayleigh noise sources to the shelf rather than to the deep sea, as observed in data from two arrays in Ireland. Second, using 3D numerical simulations in a realistic regional crustal model (including water layer and sediments), it convincingly demonstrates that this location of surface wave generation can be explained by propagation effects related to the shelf edge.

The MS then further explores the mechanisms behind the Love waves through numerical simulations in simple models. This section has great potential, but could be better explored and partly lacks clarity (see below).

Finally, the MS models the wavefield for a realistic distributed source distribution and compares observed Love/Rayleigh amplitudes and energy with simulated ones.

--

The study has a number of strong points:

- It brings interesting new insight into a complex problem for which the literature is sparse.
- It is fundamentally sound, with no demonstrable errors (although it may not fully exploit its potential).
- The 3D simulations are interesting; I have not seen anything of similar quality for continental margins.
- The subject is in itself of general interest: A better understanding of their origin gives insight into wave propagation, coupling of different wave types at the sea bottom, and can be beneficial for better imaging using Love waves present in the seismic noise.

This work should be published, whether in Nature Communications (see point 1 below) or elsewhere. My suggestions should be seen in the light of further promoting the impact of this work, which I consider of high quality. I therefore recommend major revision, but point 1 below may need special editor attention.

Comment #1: The authors show that there is a big difference between the observed Love wave amplitudes and the simulated ones. They invoke mechanisms of why this is the case, but none of these additional contributions are quantified. Also, there is an additional problem because the fundamental mode Rayleigh waves tend to be mainly forwards scattered, with relatively small coupling to Love waves or higher modes (see for example work by Snieder, 1986, 1988). Due to the mismatch between observations and simulations, the manuscript does therefore not carry a full answer to the enigma of the strong Love waves. This was also the case for the 'continental' study by Pedersen et al., cited by the authors. The MS rather

confirms and quantifies a mechanism in wave propagation that seismologists know in general terms, and a mismatch with observations which specialists in surface wave studies have been frustrated for some time, previously in the context of ‘tectonic release’ of nuclear explosions, but now also in the area of seismic noise studies.

We thank the reviewer for this comment and agree that this was a weakness. We feel that we have resolved these issues and have made substantive modifications to both our simulations and our analysis, to address the reviewer's concern. A key point is to get the simulations to a point where they are realistic enough to capture the major features of interest. This required further modifications to both the geological model and the microseism source model that we inject into the numerical seismic wave simulations.

(i) In the first version of the manuscript, we mentioned two mechanisms were responsible for Love waves generation: Love waves generated at an interface right below the source and Rayleigh to Love wave conversions, the later bringing a more limited contribution. Thanks to the reviewer's comment, we realised that this distinction was not clearly stated and have improved it by better quantifying the contribution of both mechanisms. This is done by investigating the detailed wavefield associated with each mechanism (Fig. 4 and 5) and by considering both mechanisms for L/R ratios calculations (Fig. 6), to highlight their respective contributions.

(ii) We have also made substantive changes to our regional synthetic simulation (Fig. 8):

Synthetic model of the Irish offshore: In the first version of the manuscript, we restricted the sediments to a minimum Vp velocity of 2.5 km/s due to limitations in the mesh resolution. In the revisions to the paper we have changed the velocity model and remeshed the model to be able to include similar resolution for slower sediments (2 km/s), which is more realistic based on local seismic velocity studies. We have run a whole new suite of simulations in this higher resolution model, with more realistic near sea floor sediment structure.

Source for the “extended source” simulations: We now consider the full secondary microseism frequency spectrum derived from the ocean wave model and available through the P2L data (Fig. 8). We take the median of the P2L data over a year (03/16-03/17) to generate a median P2L frequency spectrum for each geographical point, discretized here with a 12.5 km grid spacing. Through power spectrum interpolation, the P2L spectrum at each grid point is transformed into a continuous acoustic signal with random phase. This leads to a multi-point spatially distributed acoustic source, continuous in time and containing the full spectrum of the secondary microseism excitation over the area defined by the model. More explanations can be found line 396-409 and in the method section line 624-642.

We are confident that the synthetic data are now even more realistic, following these modifications.

(iii) Finally, we have also changed the way we calculate the Love to Rayleigh (L/R) ratios from both real and synthetic array analysis (description at line 659 in methods section). Whereas previously we considered only the overall dominant azimuth (extracted from probability density maps (Fig. 1)) into which we rotated the data for the full time period considered in the analysis, we now consider each time window defining the array analysis process, separately. This means the L/R estimates from synthetic and real array analysis are

calculated for windows of 3600s for the real data and 200s for the synthetic data. Vertical and transverse channels were beamformed based on the back azimuth and slowness associated with the maximum beampower for each time window. This enables us to obtain L/R ratios distributions for both synthetic and real data array analysis (Fig. 9) which we feel are more realistic than previously.

To make the study of more general rather than specialist interest, other areas for which Love wave source locations have been identified could be included (but this would be a huge work) to check whether the conceptual model presented by the authors works elsewhere. The authors could also further quantify the additional effects that are presently just mentioned as a potential explanation to the mismatch between observations and simulations.

If the hint of Love wave generation being efficient only when the source is located above a strong heterogeneity (see below), it should be possible to relate changes in transverse energy with high waves in very specific areas. This would probably be a frequency dependent effect. If that type of geographical effect was demonstrated, that would bring a very interesting insight into Love wave generation for seismic noise.

While in the present stage I am not certain the manuscript is sufficiently general to warrant publication in Nature Communications, I would recommend to give the authors a chance to further improve and generalize the study.

We appreciate the reviewer's comment. We think that this is a key point and we have added a substantial amount of additional material to address it in order to make the study more general. As hinted at by the reviewer, building detailed models for other regions would be an enormous job (equivalent to replicating this study several times) so we have taken a different approach with simplified generic models, which we feel still generalises the main points of the study (see comment below). As mentioned above, first we re-worked our model of the Irish Offshore to include more realistic low velocity sediments as observed from local seismic studies. We also modified the ocean source that is injected as the input disturbance for the numerical seismic wave simulations. The combination of these changes has substantially improved the comparison between synthetic and real L/R ratios estimates. This suggests to us that, in principle, the origins of the real Love waves are likely captured by the mechanisms proposed in this paper, but that their details (e.g. absolute and relative amplitudes to Rayleigh waves) depend on the details of both the source and structural model. However the details of real source and real structure are difficult to determine, but we feel that we have invested substantive effort to capture both as best we can, in order to render the comparison between real and synthetic L/R ratios more informative in terms of unlocking the origin of Love waves.

One could generalise the results by building detailed geological models and running full 3D simulations and complex local ocean sources in other geographical areas around the world (provided that local geological information was available in sufficient detail). However reviewer #1 rightly identified this as a 'huge task' which we feel is not within the scope of the study as we would effectively have to replicate the work here in multiple locations. Instead we have opted to use simplified models that capture the essential features present in other locations, that we have identified as important in the detailed model, offshore Ireland. This allows us to use these simplified models in a generalised way. To do so, we have defined a 3D

concept model (Fig. 4a) where we vary water depth, sediment thickness and velocities, in order to reflect the structural configuration of multiple geological environments around the world. All those parameters are now discussed more in detail (line 273 to 345), including how they affect the L/R ratios (Fig. 6). In addition, we expand the discussion (line 447 to 478) on how the different synthetic L/R ratios generated for multiple structural configurations relate to the real ratios observed in different regions of the world: Continental Europe, Australia, Japan and Western US (Fig. 10).

Comment #2: My second comment is related to the simple models, presented in Figures 4 and 5 and associated text. As I understand it, their scope is to understand the mechanisms of the Love wave generation, but this section needs quite some improvement to fully enlighten the reader.

As I understand the text, the idea of the simple cases is to achieve an in-depth understanding of the wavefield. I have spent a few hours trying to relate specific arrivals in the seismograms of the figures with specific propagation paths and/or wave conversions, but so far I was only partly successful. Some of the comments and questions could potentially be addressed through snapshots of the wavefield in supp. Mat. or for example through more clear explanations in the text, identification of different wavetrains etc.

We agree some improvement was needed to clarify the different wavetrains. We hope the new Figures (Fig. 4 and 5) define the wavefield more clearly now. To do so, as mentioned above we have redefined the simpler models incorporating them as part of a new concept model (Fig. 4a) and running new suites of simulations. The aim is to highlight the different mechanisms associated with Love wave generation more clearly. What is different is that we now show data along profiles to characterize the evolution of the wavefield as it propagates. We also include dispersion analysis to better define the wavefield and, as suggested by the reviewer, also show detailed snapshots and animations for four particular case studies (Fig. 4 and 5). The animations can be found in Supplementary Figures S7, S8, S10 and S11 and more detailed dispersion analysis of the wavefield in the sedimentary basin can be found for different models in Supplementary Figures S9, S12, and S14-S17.

Below I list some of the immediate questions and suggestions that spring to mind, but there are surely other key information and conclusions that could result from more in-depth analysis of the 'simple' cases.

a. For me, one of the most interesting observations is that the Love waves seem to be present even in a very simple geometry (bathymetry part of fig 4), if the source is located in a laterally heterogeneous area. This is clearly illustrated by the case S1-R1. This idea may possibly be a key ingredient for the Love wave amplitudes. Indeed, looking at the case S2-R1, the Love wave seems to be minimal (waves around $t=40s$, if I got distances and velocity right). This would mean that the case S1 is more favourable than S2. Is there any way to explore this idea further?

Although a more complex bathymetry might well lead to particular focusing and constructive interferences of Love waves, increasing the amplitudes, we did not investigate that part in

details. However, by simply changing the structure (water depth, sediment thickness and velocities) we are hoping to be characterizing the changes in the wavefield amplitudes in a more detailed way now (Fig. 6). We now show via Figure 5b and Supplementary Figure 13 that the presence of dominant Rayleigh higher modes contributes more efficiently than the Rayleigh fundamental mode to Love wave conversions at the shelf break. This part is discussed at line 260. (See next comment)

b. Again, if I got the times right: what is the large second wave train? Does it correspond to the slowly propagating wave on the left side of S1 in the snapshot? If so: could the high amplitudes on the transverse component for that wavetrain be due to a simple change of propagation direction due to change in wave speed ? (especially as it seems to arrive at the same time as the waves on the vertical and radial components). Which arguments do the authors have to say that the late arrival on the transverse component is really a Love wave? I don't think the timing would be explained by different modes, for example, but if it is the case, the MS needs to have a clearer and quantified explanation, with information on higher mode wave velocities at the dominant frequency.

Thank you for your comment. We now study this effect through Supplementary Figure S13. We have also added dispersion analysis for all simple simulations to quantify better the wavefield and associated phases, in particular to differentiate Love waves. Through the 3D concept model (Fig. 4a) we have also extended the scale of the observations enabling bigger source/"conversion points" - receiver ranges for more clarity on the wavefield evolution.

On Supp. Figure S13, we can see the pseudo-Rayleigh fundamental mode interaction with the shelf leads to Rayleigh wave velocities on the transverse component. We explain that this is due to pseudo-Rayleigh fundamental mode acoustic to elastic conversions at the shelf break that seem to leak onto the transverse component (line 263). In contrast, being mainly dominant in the elastic domain, the first overtone pseudo-Rayleigh mode shows clear Love wave conversions (Supp. Fig. S13).

The effect is not clearly seen in Figure 5b since, due to the presence of the sediments, the Rayleigh wavefield is mainly dominated by higher modes that will lead to Love wave conversions at the shelf break.

c. The same observation seems to hold for the Sediment case, where the S1-R1 case has a clear and early arrival on the transverse component (even though the vertical and radial scales are different from that of the transverse component – I would btw suggest to have identical vertical scales on all components, considering the question at hand).

As mentioned just above we hope that Figure 5b and Supplementary Figure S13 helps demonstrate better now the conversions of the different Rayleigh modes as they interact with the shelf break. We are also now keeping the same scale for all components throughout all Figures.

d. For this case, the S2-R1 case seems to have early energy on the transverse component (attention vertical scale) for the late arrivals, but hardly for the early ones. What is the interpretation of this?

We show now that for such water depth and sediments velocities, the wavefield is actually quite complex (Supp. Figure 9). Multiple modes are mixed for Rayleigh and Love waves which explains such differences on the arrival times.

e. The case of S3 is interesting: the source is in a very heterogeneous area. The Rayleigh waves have a very long duration, but there is significant energy on the transverse component. However, in terms of energy, the increase of duration of the Rayleigh wave means that the total energy on the transverse component is very small as compared to the vertical and radial components. Are there other frequency intervals where the transverse component demonstrates more resonant behaviour in the sedimentary basin? Could such resonance effects be of interest with Love wave generation? What if the lateral heterogeneity coincides with a depth of resonance for the Love waves?

Thank you for your comment. We find the wavefield to be quite complex and therefore do not discuss further Love wave resonance in detail. However, we consider multiple models to focus more broadly on the structural control over the L/R energy ratios on Figure 6.

f. In figure 5, it is necessary to explain which wavetrains were used for the analysis, because of the issues with the different wavetrains discussed above.

We now show that only fundamental modes for Rayleigh and Love waves are dominant on the shelf (Fig. 4-5), where the ratios are calculated (Fig. 6). Therefore, we assume the full length of the signal is valid for the use of fundamental mode eigenfunctions for the L/R ratio calculations which are now shown on Figure 6.

g. Also in Figure 5: Are the variations between azimuths and models more related to increase/decrease in R&Z amplitude or increase/decrease in T amplitude, or a combination of both? Between models 1 and 2: are the differences related to change of resonance frequency of Love and Rayleigh waves in the basin?

We now show the RMS amplitudes associated with vertical and transverse components for multiple models on Figure 6. We discuss that although we observe a structural control over the amount of Love waves generated in the ocean, it is really the geological environment influence on ocean generated Rayleigh waves that will help modulate the L/R ratios on land (discussed line 300).

Minor comments:

Comment #3: Distances could be shown in km rather than m. It would be nice to know the precise coordinates of sources and receivers; it is quite difficult to figure out based on the figures (or did I miss something ?)

We now express all distances in kilometres and have added range values on Figure 4 and 5 for a better idea of the source-receiver distances.

Comment #4: We should be informed of the propagation velocities of Rayleigh and Love fundamental mode waves in each part of the two models, for the dominant frequency (table would be fine). This would make it more easy to relate wave arrivals with specific waves.

We now provide more details on the Rayleigh and Love wave velocities for the modes present in deep water, the sedimentary basin and shelf area. This is done through dispersion analysis and a comparison with dispersion curves from 1D theory showing the different modes

expected for Rayleigh and Love waves. Those comparisons can be seen on Figures 4, 5 and 7, and Supplementary Figures S9, S12, and S14-S17.

Comment #5: I would suggest to change numbering between R1 and R2. This means that the numbering would correspond to a progression in discussion from the simplest case (presently S1-R2) towards increasingly complex cases.

We have changed the models and station configurations and run a suite of new simulations compared to the previous version of the manuscript, and hope that the whole new layout and additional information brings more clarity.

Comment #6: For figure 5, I would suggest that the authors delete model 3. It is a useful test, but can be mentioned in the text only, and it complicates the figure unnecessarily.

We agree and have removed it. For simplicity, we do not include intrinsic heterogeneity in the models anymore as it has been already discussed in detail by Ziane, D. & Hadziioannou, C. 2019 (mentioned in the text, line 430). Instead we focus only on the structural control of the sediment and water layers over the L/R ratios.

Comment #7: I really appreciate the layout and content of Figure 7

Thank you, we keep the same layout, now in Figure 9, in the revised version of the manuscript although the L/R ratios calculation method have been revised (see comment #1).

Comment #8: Only the hours where the backazimuth is defined within $\pm 10^\circ$ of the dominant beam direction over the full year were considered. It is not easy to understand the impact of this choice: so we are looking at Love waves that are the closest of the situation in the simulations? A clarifying sentence would be helpful.

Thank you for your comment. It helped trigger us to change the way we calculate the L/R ratios from the arrays (see comment #1) and overall, we think our approach is now both clearer and more appropriate.

Comment #9: Supp fig 4 does not need to have the vertical axis go as far as 3. Two would be sufficient.

This figure has now been removed but we took that comment into consideration for other pictures.

Comment #10: Supp fig 6 : please give definition of propagation direction (I assume it is not clockwise angle from north of the incoming wave).

We have now updated the array analysis pictures and replaced propagation direction (azimuth) with back azimuth values. The plots for all array analysis can be found on Supplementary Figures S1-S3, S5 and S21-S23.

Comment #11: Double check formulation and text for all figure captions, there are a couple of misses and bad sentences.

We have checked the text and made some corrections.

Comment #12: Some figure text is too small

We agree and have increased the size of the text on the figures.

Comment #13: Potential references to add or have a look at:

- Approximation of surface wave mode conversion at a passive Continental margin by a mode-matching technique, T.Meier and P.G. Malischewsky, GJI 2000 : there are so few papers on the simulation of Rayleigh/Love waves at continental margins that the authors should include this one.
- S-to-Rayleigh wave scattering from the continental margin observed at USArray, Buehler J.S, Mancinelli N.J, Shearer PM, GRL 2018 could be of some interest?
- Imaging lateral heterogeneity in the northern Apennines from time reversal of reflected surface waves, Stich et al., GJI 2009 : also has additional insight even though not from continental margin.
- Influence of the seismic noise characteristics on noise correlations in the Baltic shield, Pedersen et al., GJI 2007: supports the results of the present study, as it shows that the 2nd microseismic peak Rayleigh waves may be distributed along the continental margin, as compared to a more focussed noise direction in the first microseismic peak.

We would like to thank the reviewer for the suggestions. On body wave to surface waves scattering, instead of Buehler et al., 2018 and Stich et al, 2009, we have now added Yu et al., 2017 (line 211) which discusses teleseismic SH to Love waves scattering in the continental margin context.

Yu, C., Zhan, Z., Hauksson, E. & Cochran, E. S. Strong SH-to-Love Wave Scattering off the Southern California Continental Borderland. Geophys. Res. Lett. 44, 10,208-10,215 (2017).

We also added Pedersen et al., GJI 2007 (line 154 and 473) for secondary microseism surface wave sources observed in coastal/shelf areas and in particular offshore the coast of Norway.

In addition, we now mention T.Meier and P.G. Malischewsky, GJI 2000 (line 163) for mode transmissions and conversions associated with continental margins.

#####

Reviewer #2:

Le Pape and co-authors report on a study of the generation of Love waves in the secondary microseism. Although many observations have shown the presence of Love waves in the microseismic wave field, no clear generation mechanism has been found so far.

Here, the authors use numerical simulations show that conversions from Rayleigh waves can result in microseismic Love waves, in the case of some specific bathymetry and subsurface configurations inspired by the regions offshore of Ireland.

By simulating a few examples, the authors show that factors such as bathymetry, sediments and the geometry thereof has an effect on the amount of Love waves generated. The analysis is not very systematic and the interpretation stays mostly qualitative, but it is enough to show that the composition of the transverse polarized surface wave field can be a complex interplay of many factors.

This confirms what the literature about the secondary microseism has been suggesting so far, and it is one of the few studies that uses forward modeling in their investigation. In that sense, the results are of interest to the scientific community. The manuscript contributes to our understanding of how the composition of the secondary microseismic wave field emerges.

The manuscript presents a lot of interesting results (such as, for example, the strong azimuthal dependence of L/R ratios, the complexity of the wave field in the presence of sediments, etc), which unfortunately are not discussed in depth. This is likely due to constraints on the length of the article.

In my opinion, the paper would also benefit from a bit more discussion of the relative contribution of different factors (sediments, bathymetry, source) to the excitation of Love waves.

The paper is well written and the figures present the results well enough. A few more figures could be included in the supplementary material to highlight some interesting phenomena.

Two main things raise some questions about this manuscript:

Comment #1: the assumptions about the source locations, and subsequent comparison of simulation results, are based on observations of the secondary microseism in Ireland. Not enough information is given in this manuscript to render these observations reproducible. E.g. no information is given about the arrays used (geometry, instrumentation, ...). Beside a general description of the beamforming method, not much is detailed about the processing of the three-component array data.

So based on the information given in this manuscript, it is not possible to evaluate the reliability of these observations, which makes it difficult to evaluate how well the simulations presented here are able to reproduce real-world observations.

I suspect that the observational study is presented in a different manuscript which appears to be still under review. This 'companion' paper [Craig & Bean, G3] is cited a few times, with statements that seem to support the results presented here:

l.80: "Those observations cannot be explained by ocean wave models only, without taking into account seismic wave propagation effects [23].

l. 144: "The simulation results are very consistent with real array findings where body waves can be seen in deep water [23] and surface waves locate at or near the shelf break (Fig. 1)"

One wonders why the authors did not merge the two publications. It seems that the results presented in this manuscript depend strongly on the results presented in [Craig & Bean].

[23] Craig, D. & Bean, C. J. Double Frequency Microseism Sources Observed with Seismic Arrays and an OBS Network in Ireland. *Geochem. Geophys. Geosyst.* __in review__, (2019).

We appreciate the reviewer's comments which have helped us improve the manuscript. We have now added more details on the array analysis and processing in order to make our analysis more independent of the manuscript of Craig et al. 2019 (still under review). Specifically:

(i) We have now included more descriptions on the instrumentation and processing for the array analysis in the methods section, including information on the array data availability for the analysis period, arrays geometry and response functions, see Supplementary Figures S25-26.

(ii) We added more details for the array analysis of all different datasets, in the method section. These include: (a) real data analysis, analysis of synthetic data from (b) the point source and (c) “extended source” simulations. For all three analyses we have added details for each period bands on the back azimuth, slowness and semblance that can be found in Supplementary Figures S1-S3, S5 and S21-23. We are also using exactly the same frequency bins for the real array analysis and the “extended source” synthetic analysis for consistency.

(iii) We have introduced more explanations in the method section for the generation of the probability density maps (line 557), supported by Supplementary Figures S27 and S28.

(iv) We have also reformulated some of the sentences with strong dependence on the Craig et al. (now ref #27) results, aiming to generalise their context.

on the 1.80 comment: Now line 84 *“With secondary microseism sources in the North Atlantic expected to be more dominant south of Greenland and Iceland^{26,28}, the observations from the Irish arrays cannot be explained by ocean wave models alone, without taking into account seismic wave propagation effects²⁷.”*

on the 1.144 comment: Now line 152 *“The simulation results are consistent with real array findings where body waves seem to mainly originate in deep water^{42,43} while surface waves are more often located near the coast and shelf break^{20,21,27,44}.”*

Comment #2: in figure 4, the arrival times for some of the transverse components are confusing. In 4c, for the S1-R1 path, the transverse component arrives before the radial and vertical component arrivals. However, for S2-R2, all three arrivals seem to share the same traveltimes. I would expect the transverse arrival to come visibly earlier than the other two, in the case of a true Love wave generated either at S2 or converted at the slope (at a distance of >50km).

In figure 4e, it can similarly be noted that all three components for the S2-R1 path share the same travel time.

This raises the question whether these arrivals are 'true' Love waves, or perhaps something like a quasi-Rayleigh wave? Perhaps a Rayleigh wave from a different azimuth than assumed when rotating the signals? I could imagine such perturbations of the polarization happening in the presence of a sedimentary basin or a steep slope.

This could be investigated by considering the wave field snapshots or animations. Alternatively, using arrays of receivers in the simulations would allow to distinguish the surface wave polarization and the corresponding slowness. The authors could either use the f-k analysis presented in the methods section, or an approach such as presented in Gal et al., 2018 (cited).

Thank you for your comments and suggestions. We have now redefined the simpler models and incorporated them as part of a new concept model (Fig. 4a). We hope that this more clearly highlights the different mechanisms associated with Love wave generation. Through the new 3D concept model (Fig. 4a) we have now extended the scale of the observations enabling bigger source-receiver ranges, which helps in the tracking of arrivals. We also include dispersion analysis to better characterize the wavefield and show snapshots and animations for four particular case studies (Fig. 4 and 5). The animations can be found in Supplementary Figures S7, S8, S10 and S11, and more detailed dispersion analysis of the

wavefield on the sedimentary basin can be found for different models in Supplementary Figures S9, S12, and S14-S17.

Concerning the observed quasi-Rayleigh waves on the transverse component, we look now at this effect in detail in Supplementary Figure S13. Indeed, we can see that the pseudo-Rayleigh fundamental mode interaction with the shelf leads to Rayleigh wave velocities on the transverse component. We explain this as due to pseudo-Rayleigh fundamental mode acoustic to elastic conversions at the shelf break that seem to leak on the transverse component (line 264). In contrast, being mainly dominant in the elastic domain, the first overtone pseudo-Rayleigh mode shows clear Love wave conversions (Supp. Fig. S13).

The effect is not clearly seen on Figure 5b since, due to the presence of the sediments, the Rayleigh wavefield is mainly dominated by higher modes that will lead to Love wave conversions at the shelf break.

Comments on main text and figures:

l. 31: "The occurrence of Love waves in recorded primary microseism signals has been explained by the interaction between propagating ocean wave and sea-bottom topography" Also consider the following two papers:

Fukao, Y., Nishida, K., & Kobayashi, N. (2010). Seafloor topography, ocean infragravity waves, and background Love and Rayleigh waves. *Journal of Geophysical Research: Solid Earth*, 115(B4).

Ardhuin, F. (2018). Large scale forces under surface gravity waves at a wavy bottom: A mechanism for the generation of primary microseisms. *Geophysical Research Letters*, 45(16), 8173-8181.

l. 45: "with observed Rayleigh to Love wave energy ratios ranging from 0.4 to 1.2 or as low as 0.25."

Additional ratios are presented here:

Tanimoto, T., Lin, C. J., Hadziioannou, C., Igel, H., & Vernon, F. (2016). Estimate of Rayleigh to Love wave ratio in the secondary microseism by a small array at Piñon Flat Observatory, California. *Geophysical Research Letters*, 43(21), 11-173.

Thank you for the three additional references suggestions. We have now added them at lines 34 and 47.

l. 54: "However, the source locations derived from ocean wave models often do not agree with actual source locations determined from land-based seismic observations." References?

Actually, we realized this sentence is maybe too strong, since most land studies on source locations around the world will still show some good correlation with ocean wave models. We have reformulated it, now at line 55: *"Although, the source locations derived from ocean wave models show broad agreement with locations determined from land-based seismic observations, some differences are observed^{8,10,12}."*

Figure 1: What is the configuration of DA and GA arrays? How is the probability density obtained from the array analysis? What is the influence of the array response on the extent of

the probability density maximum? How variable is the source location over the course of the year?

More details are now presented in the supplementary information regarding the array processing for each frequency band, including back azimuth slowness and semblance estimates for the full year. Therefore, the evolution of the source locations over the year can be found for all three-components on Supplementary Figures S1-S3. Data availability at each station is shown now on Supplementary Figure S25 and the array geometry and response functions are shown in Supplementary Figure S26.

l. 73-75: "One significant advantage of these arrays is their location along the Irish coast on the North East (NE) Atlantic seaboard. There is no ‘contamination’ along the terrestrial seismic wave propagation path from inland structures"

Do you expect any effect of the cliffs along the Irish coast? As suggested by Friedrich et al. 1998 (cited), steep coastal structures could contribute to the generation of Love waves.

It is indeed quite likely that ocean wave reflections from cliffs along the Irish coast will contribute in the secondary microseism generation on the shelf offshore Ireland. However here we meant more that as the arrays are close to the coast line, major inland path effects or significant seismic wave conversions along the path, are not likely to affect the recorded wavefield. Therefore, only “oceanic path effects” will affect the data at the arrays.

As we understand it, in Friedrich et al. 1998, the mechanism involving steep coastal structures in the generation of Love waves is mainly suggested as a contribution for Love wave generation in primary microseisms. However, maybe for very short period ocean waves (<10s), some Love waves interpreted to belong to the secondary microseism might actually be linked to the primary microseism through this mechanism instead.

l. 115-118: "We start by considering a single point source located 15m below the sea surface and defined by a Gaussian wavelet source time function of duration 3s, leading to a relatively flat spectrum over the secondary microseism periods"

The source mechanism employed in the first simulations seems to be an acoustic point source (explosion?) at the top of the water layer. Generally, for the secondary microseism, a vertical pressure source (as proposed by Longuet-Higgins, 1950, cited) is assumed. I suggest that the authors argue why their choice of source mechanism is reasonable.

Yes this wasn't clear. In order to clear this point up, in our revised simulations in the new version of the paper we now use a vertical pressure Ricker wavelet with dominant period of 5s. In addition, in the simulation with the “extended source”, the P2L derived acoustic signals are injected at each P2L grid point as a vertical pressure.

Figure 3: please specify what the green star represents.

We have now added more explanations in the caption. The star represents the original source location in the simulation.

l.181: "Both Figure 4d and 4c show that the radiation pattern" is figure 4b meant?

We meant here more the radiation pattern directly affects the recorded signal for different receivers. However, the original Figure 4 has now been replaced by the new 3D concept models mentioned above.

l.183: "The transverse wave field radiation patterns are significantly affected by the source location, the lateral geometry of the shelf edge and changes in its slope steepness (Supp. Fig. 2)."

I disagree: from Supp. Fig. 2, the two values of slope steepness tested barely affect the radiation pattern, and only changes the amplitude of the resulting transverse wave field.

Reflecting on this point we agree with the reviewer. We have reworked this part in the manuscript. In the revised manuscript, we focus in more detail on the effects of the slope of the bounding edge of the sedimentary basin on vertical and transverse amplitudes, in supplementary Figure S20.

l.193: "Similar to the "bathymetry-only" model (Fig. 4a), the shape of the sediment edge will also significantly affect the radiation of the transverse wave field and therefore the Love wave energy detected by two different receivers."

This is not shown anywhere. Such a statement should be backed up with simulation results, even if only in the supplementary materials.

We agree with the reviewer. We now provide more details on the wavefield for all four cases studied (Fig. 4 and 5). Details on wavefield propagation can found in Supplementary material S6-S8, S10 and S11 where the effects of both bathymetry and sediments can be seen.

l.201: "Heterogeneity is introduced by randomly adding ± 0.2 km/s on the velocity values of model 1."

Is this value based on real velocity models? What is the scale of the heterogeneity (especially compared to the wavelength)? Is the heterogeneity self-similar, as is the case for many earth materials?

Thank you for your comment. As it is already well described in Ziane, D. & Hadziioannou, C. 2019, we have discarded the inclusion of random heterogeneities in the models to try to keep things more simple. The wavefield is already very complex in the layered sedimentary/crustal models, and we also have better control on structures in these models. The random heterogeneity in the previous manuscript version was not based on any particular study in the area (there isn't one) so we did not have the correct correlation length or fluctuation strength. In fact, considering intrinsic scattering in a heterogeneous oceanic crust for a wavefield of period 5s, Ziane, D. & Hadziioannou, C. 2019 obtained a Love to Rayleigh ratio of 0.1, which is not negligible but only represent a fraction of the ratios observed around the world. Here we now focus in more detail on the structural control of sediments and water layer on the L/R ratios (Fig. 6).

figure 5: The results presented in figure 5 are interesting, yet are hardly discussed. As cited in the introduction, L/R ratios are known from observations in other studies. The authors could at least comment on how their values compare.

Moreover, Gal et al. 2017 (cited) extensively considers the influence of sediments on the secondary microseism wave field. The authors could discuss how their results relate.

We have defined a new 3D concept model (Fig. 4a) where we vary water depth, sediment thickness and velocities. All those parameters are now discussed in more detail (line 273 to 345) and on how they affect the L/R ratios (Fig. 6). In addition, we expand the discussion (line 447 to 478) on how the different synthetic L/R ratios generated for multiple structural configurations relate to the real ratios observed in different regions of the world (Fig. 10), including Australia and the observations from Gal et al. 2017.

1.231: "used as a time synchronised input source grid for the 3D numerical simulation": I assume you are using the same point sources as before at each grid point?

In the previous manuscript version, the point sources were randomly distributed in space and time in order to define a random phase. However, we have now redesigned the source to be more realistic, and ran a completely new simulation. We now consider the full secondary microseism noise spectrum derived from the ocean wave model and available through the P2L data (Fig. 8). More explanations can be found line 396-409 and in the method section line 624-642.

1. 278: "as well as a more heterogeneous models" singular or plural?

This has been corrected.

figure 6a: the way the different periods are presented, the "strongest ocean source area" in the NW (1.237/238) is hidden, unfortunately.

We agree with the reviewer and have corrected it now through the implementation of our new source approximation (Fig. 8).

1.304-307: "We demonstrate that multiple parameters will affect those Love wave levels... and source distribution originating from ocean waves interactions." Where is this shown?

Those aspects are now discussed in more detail in association with Figure 6. The discussion is now between lines 273 and 345.

1.312: "implying that scattering along the path can enhance the "initial source values" that we see closer to the ocean." This is supported by Ziane, D. & Hadziioannou, C. 2019 (cited)

We have added the reference to Ziane, D. & Hadziioannou, C. 2019, now line 497.

Comments on the Methods section:

Not enough detail is provided for the study to be reproduced, in particular the signal processing is not fully explained. How is the back-azimuth, slowness etc obtained separately for radial and transverse components in the array analysis? How are the probability density maps obtained?

We agree with this comment. We have now included more details on the processing for the array analysis in the methods section, with clear separations of the differences in the analysis for the real data, the point source and "extended source" simulations.

For the separation of the radial and transverse components in the array analysis, we refer to the *Geopsy* software (line 523) and added the three-component beamforming implemented in *Geopsy* is performed separately for each component by rotation of the horizontal components following the approach of Poggi & Fäh (2010).

We have also added a section in the methods for the description of the probability density maps generation (now line 557).

1.382-383 please define the symbols used in the equation.

We have now described the symbols associated with the L/R ratios equation, now line 648.

Comments on the supplementary material:

figure 4 & 5: since the radiation pattern (azimuthal distribution of ratios in figure 5) and signal structure (waveforms figure 4e) seem to be so complex for the model with sediments, I think it would be very interesting to show a snapshot or animation of the wave field, preferably from source S3. This could be included in the supplementary material.

We agree. As mentioned above, we have now included snapshots and animations for all simple simulations. Those can be found in Supplementary Figures S6-S8, S10 and S11.

supplementary figure 5: please specify what the different colored lines (green, red) represent.

The figure has been changed and we now tried to avoid the use of green and red on the same plots.

supplementary figure 6: what is meant with propagation direction? Backazimuth, or backazimuth + 180 degrees?

We have now replaced the propagation direction with back azimuth values in all pictures (Supp. Fig. S1-S3, S5 and S21-S23).

Comment on figures, general:

To keep your figures accessible to color blind users, please consider using appropriate colormaps (e.g. no rainbow colormap) and differentiating lines with line styles in addition to color.

We appreciate the reviewer's comment. It is a very important point and we should have been more careful about that. We have now made some changes and hope it has improved the figure accessibility for colour blind readers.

#####

Reviewer #3:

This paper shows possible excitation mechanisms for Love waves of secondary microseisms, which are mainly related to bathymetry and sediment structure in the ocean areas, using both large-scale three-dimensional numerical simulations of seismic wavefield and observed data in Ireland. The demonstration of Love wave generations from the numerical simulations and observations are reliable and interesting, and this paper has an important contribution to our understanding of excitation mechanisms for secondary microseisms. I recommend publication with some revisions, as discussed below.

Comment #1:

- 1) On multiple modes of surface waves in the synthetic calculations

First of all, it is better for readers to define the observed signals (Fig. 4). In particular, there are two signals in the right panel of Fig. 4c, while one pulse can only be seen in the left two panels of Fig. 4c. The authors stated that those two waves are the fundamental and higher mode (L180), but it seems that more detailed descriptions are required, as shown in Fig. R1. In addition, I do not understand wave propagations in the T component for S3-R1 and S3-R2 (Fig. 4e and Fig. R2), because their travel times are significantly different from the Rayleigh wave ones. Please explain more details on the propagations.

(Suggestion 1) I recommend that the authors add some figures showing record sections (waveforms aligned as a function of distance) at each component for S3-R1 and S3-R2 with putting more receivers, e.g., like light-blue and red circles in Fig. R2. It is expected that these plots show propagation velocities of the observed signals and also the conversions of surface waves at the slope (continental shelf), particularly for Love waves. If possible, such record sections for the case in Fig. 3 are also helpful for readers.

Thank you for your comment. We have redefined the original simpler' models to be incorporated as part of a new concept model (Fig. 4a) to highlight the different mechanisms associated with Love wave generation. We like the reviewer's suggestion and hope the wavefield evolution is clearer as we are now showing data along profiles to track the evolution of the wavefield as it propagates. We have also included detailed snapshots and animations for four particular case studies (Fig. 4 and 5) that can be found in Supplementary Figures S7, S8, S10 and S11. We also include dispersion analysis to characterize better the wavefield and mode conversions. The dispersion analysis is performed for the wavefield on the shelf and deep-water areas (Figures 4 and 5), as well as for the sedimentary basin (Supp. Fig. S9).

As it is relatively complex, we did not include a profile section for the case in Figure 3. We feel that the new concept model and associated simulations (Figure 4 and 5) are now sufficient to describe the propagation of the wavefield.

Secondly, one of the important findings of this study is that, in Fig. 4, the two Rayleigh waves have comparable amplitudes, which have not been observed in two-dimensional numerical simulations in previous studies. Can these features be seen at other propagation directions for Love waves in the simple models or the complex 3D model? If yes, when estimating L/R energy ratios, the amplitude balances of these two waves are very important. (I mean that, for some geometries between sources and receivers, the former amplitudes of the Rayleigh waves in the Z component are larger than the latter amplitudes, while the former ones of the Love waves in the T component are smaller than the latter ones, and vice versa. Such amplitude balancing may affect the estimation of L/R energy ratios.)

This is an interesting question. We now show that the different Rayleigh wave phases observed on the shelf represent the Rayleigh fundamental mode and higher modes convert at the shelf to fundamental Rayleigh wave. Such an observation was already mentioned in Gualtieri et al, 2015 (cited).

Gualtieri, L. et al. On the shaping factors of the secondary microseismic wavefield. J. Geophys. Res. Solid Earth **120**, 6241–6262 (2015).

A similar conversion happens for the Love waves at the sediment basin boundaries as seen now in Figure 4c, which shows Love wave's fundamental mode also dominates on the shelf

(Fig. 4c), despite the presence of Love wave's higher modes in the sedimentary basin (Supp. Fig. S9).

As seen on the dispersion analysis (Supp. Figure S9), the wavefield is quite complex with multiple modes crossing each other. Although it would be very interesting, it is quite difficult to properly assess the amplitude contribution of a particular mode as a function of the propagation direction.

(Suggestion 2) *This is just based on my curiosity, so the authors do not need to reply this. When heterogeneities in model 3 (Fig. 5) have anisotropy in correlation lengths between the x and y axes, are the radiation patterns of Love waves and their higher modes different? I just suppose that, if there are some mechanisms that change the radiation patterns of the modes, it is very interesting and possibly affects L/R energy ratio estimations.

That is an interesting point that would definitely need further investigations. However, at the end we discarded the introduction of heterogeneities in the models to try to keep things simpler as the wavefield is already very complex in the layered models.

Comment #2: On seismic frequency ($2f$) and ocean wave frequency (f)

Secondary microseisms are excited by wave-wave interactions of ocean swell, but the secondary microseisms have double frequencies compared to those of ocean swells. The authors used P2L between 4 s and 10 s in Fig. 6, but which do those periods correspond, $2f$ or f ? It is nice if the authors explicitly mention this in the text.

We have now added in the caption of Figure 8 (line 398): *“The P2L frequencies are defined by the secondary microseism frequencies which are double the frequencies of ocean waves.”*

Minor comments:

L9–11

It is better to state that this sentence corresponds to secondary microseisms. We have specified secondary microseism on line 9.

L18

Love waves do -> Love waves observed in land do

We have implemented “Love waves observed on land...”, line 19.

L81-82

In the period of 5–8 s, the source locations estimated using the Z and T components are different, so that I confused this sentence after checking Fig. 1.

We have now specified it is particularly the case for the period band 3-5s (now line 88).

Caption of Figure 2

–L98 What is “a min period resolution”? If the sampling rate of the synthetic waveforms is 0.33 Hz (3 sec), it seems that the authors can use up to 0.166 Hz, which is lower than the frequency band of secondary microseisms.

The minimum period resolution is not related to the sampling rate but will depend on the synthetic wavelength resolution driven by the 3D mesh discretisation. We have added more explanations in the methods section for the period resolution in the simulations, line 597: *“The minimum period resolution is defined by the minimum wavelength that the 3D mesh can*

resolve in the simulation without introduction of numerical dispersion. It is therefore mainly dependent on the velocity model.” For more clarity, the sampling rate for each synthetic signal is also now described in the methods section (lines 601 and 620).

L119

located above a -> located in a

We made the change.

Caption of Fig. 3

-Please explain more details, including red regions, light-blue stars, and red triangles in Fig. 3a and red squares in Fig 3b.

-located just below the sea surface -> located 15 m below the sea surface

-Please remove “mechanism” in L131.

We have made the changes to the caption of Figure 3.

L140

Please define “elastic Rayleigh waves in shallow water”. I think that, even 300 m water depth, the observed surface wave is the pseudo-Rayleigh wave.

Yes. We meant the dominant part of the fundamental mode will be in the elastic domain. We have now added line 147 “dominant elastic Rayleigh waves in shallow water (<300 m depth) for the periods of interest”.

L142

the wavefield apparent propagation direction -> apparent propagation direction of the wavefield

We have implemented the modification.

L181

Figure 4d -> Figures 4b

Figure 4 has now been replaced.

L237

I cannot understand the location of the strongest ocean source area in Fig. 6b. Please identify it with a symbol.

We have actually redefined our ocean source for the simulation and hope the location of the strongest ocean source area is clearer now in the new Figure 8. More explanations can be found line 396-409 and in the method section line 624-642.

L249

It is better to specify the directory of P2L in the ftp site.

We have now added more details, thank you

line 632: “/ifremer/ww3/HINDCAST/GLOBAL/2016_ECMWF/p2l for the year 2016”

line 633: “/ifremer/ww3/HINDCAST/GLOBAL/2017_ECMWF/p2l for the year 2017”

Fig. 6b and L258

The snapshot of the wavefield is unclear. Please improve it.

Actually, with a total of 3097 continuous point sources associated with the “extended source” synthetic simulation (now Fig. 8), we realised that a snapshot of the wavefield does not make much sense now. It is impossible to really see any clear propagation, so we decided to remove it.

L301

I agree that a dipping interface at the bottom of sediments potentially produces Love waves within marine sediments in addition to Love waves whose energies are concentrated in the underlying crust. In numerical simulations, are such sedimentary Love waves transmitted to land? I am interested in reflection and transmission of Love waves at the sediment boundary slopes, and whether such transmitted Love waves affect L/R energy ratios. It is nice if the authors state on weak or strong transmissions of Love waves at the sediment boundary slope in the text.

This is a nice question. We now mention, line 332, how the basin edge slopes will affect the wavefield transmission but do not go into details. However, we now show the RMS amplitudes associated with vertical and transverse components for multiple models on Figure 6. We demonstrate that although we observe a structural control over the amount of Love waves generated in the ocean, it is really the bathymetry and importantly the geological influence on the ocean generated Rayleigh waves that will strongly define the L/R ratios on land.

L341

The authors only explained the method of the numerical simulations using the complex 3D structure, and did not explain the bathymetry and sediment models in details. In particular, it is better to explain the heterogeneous structure in model 3 (e.g., correlation length and other parameters, and why the authors chose them here) and the sediment structure in models 1–3. It seems that the sediment structure and the location of S3 represent the situation on the T component case of 5–8 s in Fig. 1.

We have now added more details on the numerical simulations for all our models at lines 586 to 636. However, as mentioned above, we have discarded the inclusion of random heterogeneities in the models to try to keep them better constrained and more simple, as the wavefield is already very complex in the layered models. (see comment above on this point)

On the T component case at 5–8 s for Figure 1, the probability density maps will mainly define where the back azimuth that the two arrays are pointing to but does not necessarily mean that the source is there. That is why we did not discuss in detail a comparison between the simple models and the array analysis of the real data.

L382

For the equation, please define the variables used.

We have now defined the variables in the equation, now line 648.

Throughout the paper: please unify from meters to kilometers in the text and figures.

We have now unified the units to kilometres throughout the manuscript.

REVIEWER COMMENTS

Reviewer #1 (Remarks to the Author):

The MS has improved very significantly. It is more in-depth and informative, and gives the reader the necessary elements to judge for himself/herself when reading.

Reviewer #3 (Remarks to the Author):

Review comments on Le Pape et al., resubmitted to Nature Communications

I reviewed the previous version of this manuscript, and feel that the authors genuinely addressed the comments from the three reviewers. I am almost satisfied with the current version of the manuscript, but there are still some minor concerns that should be addressed by the authors.

Throughout the paper

It is necessary that the authors cite "Methods" section at appropriate parts in the text.

L177

What about the dip angles of the interfaces between layers of 6 km/s and 6.5 km/s and 6.5 km/s and 6.9 km/s? Are these angles the same as the bathymetry slope and the edges of the sedimentary basin?

L198-269 and Fig. 6

The authors attempted to calculate wave propagations for various velocity models with changing sediment thickness, water depth, and shear velocity in the sediment, and found that the Rayleigh and Love wave amplitudes observed at land are changed by such parameters. For Love wave amplifications, can the authors compare your results with a theory from the following paper?

M. van der Baan, The origin of SH-wave resonance frequencies in sedimentary layers, GJI, 178, 1587-1596, 2009.

When large Love wave amplitudes are calculated by the authors' simulations, does the parameters used, such as sediment thickness, dominant frequency, and V_s , correspond to the eq. (1) in van der Baan (2009)? If yes, it is better to state it by citing the paper.

L211

sediment velocity of 2.5 km/s -> sediment velocity, V_s , of 2.5 km/s

Figs. S7 and 8 (L219)

It is better to state that transverse and vertical components are left and right panels, respectively, in the figure caption.

L250 and 259-260

Why can the authors identify the dominant mode conversions from the dispersion curves? Fig. S9 shows the fundamental and higher modes of the Rayleigh waves, and it seems that both modes are possibly converted to Love wave. Also, it is hard to distinguish those modes in the records section in Fig. 5a. Please explain more details on the mode conversions in the text.

Suggestion) Is it possible to identify the fundamental and higher modes of the Rayleigh and Love waves in the record section in Figs. 4 and 5? Those in Fig. S13 are very clear, but it is nice if the

authors specify them in those figures.

L275

station located on the shelf -> station (st15 in Fig. 5) located on the shelf

Fig. 6

I do not understand the right panels. What are "Bathy." and "Sed."? Please explain.

L368

gird -> grid?

L416

Although the authors state "quite consistent", it seems that T/Z and L/R in the synthetic calculations are lower than those in the observations. It is better to explain the consistency more details, or remove "quite consistent" from the text and explain possible reasons for the difference between the calculations and the observations.

L542

DA and GA should be explained in more appropriate part, because these abbreviations are used in L122 and L515-516.

L554-583

I do not understand the details in this part. The following contents are necessary to understand this part. What kind of the observed data did the authors use in this part to obtain vertical red lines in Fig. S27? Is $f(x)$ the probability density map, and is x the location? Did the authors use this technique to make Fig. 1 using the data in Figs. S1-S3?

I like your record sections for wave propagations, which allowed me to compare the three component amplitudes easily. Thanks so much for your hard works.

REVIEWER COMMENTS

Reviewer #1

The MS has improved very significantly. It is more in-depth and informative, and gives the reader the necessary elements to judge for himself/herself when reading.

#####

Reviewer #2

The authors almost addressed the comments, but some minor revisions are still needed before the acceptance. In my comments, blue and light-blue colours in the text below are used for revision-necessary parts and just my confirmation, respectively.

The reviewer was satisfied with most replies from our previous revision but highlighted some new comments (in blue in the “reviewer’s response to authors” PDF file). For clarity, we have only kept here those revision-necessary comments with answers to each of them below.

Comment #1: I recommend that the authors add more explanations in the caption of Figures S25, particularly for the meanings of blue colour and the lengths of vertical bars. I cannot understand the data availability from this figure and caption.

We agree with the reviewer that more explanations are needed. In the previous version of Figure S25, the blue and red vertical lines respectively represented overlapping data and gaps. However, since the aim here is to emphasise on the lack of data over relatively long period of times, instead of adding those explanations in the caption, we have simplified the figure by removing the apparent very short gaps and labels of overlapping data, taken into account during processing. We think the updated Figure S25 and associated caption now highlight data availability more clearly.

Comment #2: Please delete comma ‘,’ after ‘Although’ in the main text.

We have made the change. Now line 55

Comment #3: I agree that seismic wave propagations observed at an array near the coast line are not significantly affected by coastal structure, but I do not understand the last sentence of the authors’ response. Does this mean that Love waves in the secondary microseisms can be non-linearly linked to those in the primary Love waves? The frequency ranges of the two

Love waves are different, so it is necessary to consider a non-linear interaction between the two due to the presence of steep coastal structure. Please clarify the authors' response again, and also I would like to know references that show examples for such non-linear interactions.

We might have created some confusion the way we wrote the sentence. We do not infer a non-linear link between primary and secondary microseism Love waves and are not aware of any studies that do so. We have looked through the manuscript again to make sure that we don't imply this in the text of the paper. We were only proposing a possible misinterpretation of primary microseism Love waves to be secondary microseism Love waves. For example, ocean waves of 8s period interacting with a steep coast would potentially generate primary microseism Love waves (as proposed by Friedrich et al. 1998). However, those Love waves recorded at a receiver could as well be interpreted as secondary microseism Love waves, being in the secondary microseism period range.

Comment #4: I would like to see some description on the resampling interval on the sentence of "At each grid point, the P2L spectrum is resampled in order to extract pressure time series with a sampling rate high enough to match the sampling rate required by the simulation". What value is a sampling rate "high enough"? Did the authors interpolate from the sampling rate of 3 hours of P2L to a specific sampling rate?

We agree with the reviewer that point needed clarification. We have now added the following paragraph line 663: *"The time step used for the simulation is $dt=0.003125s$. It is constrained by the size of the mesh elements in comparison with the wavelength of the simulated signal in order to have a numerically stable simulation. Here, the external source defined by the pressure time-series requires the same sampling rate. Since the downloaded P2L data in the frequency domain has a coarse sampling rate (orange dots on Figure 8), it needs to be resampled in order to have a P2L spectrum with frequency resolution $df= 1/(dt*N)$, where N is the number of time steps in the simulation ($N=680,000$)."*

Comment #5: Figure 6 is interesting, and I would like to see some descriptions on the robustness of each value in Figure 6. For example, although this figure was created from RMS results at a single station (line 274), was the observed tendencies on each parameter in Figure 6 (i.e., the Rayleigh wave amplitudes are significantly affected by sediment velocities: line 300) not affected by station locations?

Suggestion) It is helpful for readers to read why the authors still select a station on the line when L/R and T and Z amplitudes are measured. In terms of the radiation patterns of generated surface waves, is the station suitable for measuring those amplitudes? If yes, I recommend that the authors state reasons on the selection of suitable stations.

We agree that taking into consideration the robustness of the values presented on Figure 6 is important and that is why we actually compared two station locations on all plots. We think there might be some confusion from the way Figure 6 and its annotations were presented that do not properly highlight that aspect. For instance, we have realized that the source location associated with all plots was not clearly stated and have modified the figure hoping it brings more clarity on how the effects of the different variables compare and in particular with different station locations. We could have considered more receivers but with each source model being associated with the visual radiation patterns described in Supplementary Figures S8 and S10, we thought considering two stations was enough here to see how the ratios can change depending on where they are measured. Some editing has also been done to the caption of Figure 6 (line 303) and surrounding text (line 286). We hope it makes the section clearer.

Comment #6: Addressed. By the way, I recommend that the authors define Z and T in the text, although these are defined in the figure captions in Figure 1 and 5. Moreover, it is better to defined them in line 229, not in line 232.

Thank you for the suggestion, we have now defined Z and T in the modified text line 209 in association with the description of Figure 4 results. We have also now defined them on line 240 at the beginning of the caption of Figure 5.

Comment #7: According to the reviewers' comment, I recommend that the authors change colours in Figures S4, S19, and S26.

We have now updated the colours on the 3 figures.

#####

Reviewer #3

Review comments on Le Pape et al., resubmitted to Nature Communications

I reviewed the previous version of this manuscript, and feel that the authors genuinely addressed the comments from the three reviewers. I am almost satisfied with the current version of the manuscript, but there are still some minor concerns that should be addressed by the authors.

Comment #1: Throughout the paper It is necessary that the authors cite "Methods" section at appropriate parts in the text.

We agree with the reviewer and have now added a reference sentence to the *Methods* section in several parts of the manuscript: now line 77, 119, 179, 288 and 384.

Comment #2: L177 - What about the dip angles of the interfaces between layers of 6 km/s and 6.5 km/s and 6.5 km/s and 6.9 km/s? Are these angles the same as the bathymetry slope and the edges of the sedimentary basin?

As pointed by the reviewer, it needed clarification, now line 186: "*In the models the crustal interfaces are defined so that they characterize layers of equal thicknesses between the base of the sediments and 20km depth.*"

Therefore, the dip angles do not represent exactly the same angles as the slope of bathymetry and edges of the sedimentary basin.

Comment #3: L198-269 and Fig. 6 - The authors attempted to calculate wave propagations for various velocity models with changing sediment thickness, water depth, and shear velocity in the sediment, and found that the Rayleigh and Love wave amplitudes observed at land are changed by such parameters. For Love wave amplifications, can the authors compare your results with a theory from the following paper?

M. van der Baan, The origin of SH-wave resonance frequencies in sedimentary layers, GJI, 178, 1587-1596, 2009.

When large Love wave amplitudes are calculated by the authors' simulations, does the parameters used, such as sediment thickness, dominant frequency, and Vs, correspond to the eq. (1) in van der Baan (2009)? If yes, it is better to state it by citing the paper.

We did not consider the theoretical quarter-wavelength equation discussed in van der Baan (2009) as we show the water column (not taken into account in van der Baan (2009)), also plays

a role in the observed Love wave excitation. We discuss that aspect through Figure 7 showing Love wave modes are getting excited differently depending on the water column thickness.

Comment #4: L211 - sediment velocity of 2.5 km/s -> sediment velocity, V_s , of 2.5 km/s
We have implemented the modification; it is actually a V_p value. Now line 221

Comment #5: Figs. S7 and 8 (L219) - It is better to state that transverse and vertical components are left and right panels, respectively, in the figure caption.

We now specify in the captions of Figure S7, S8, S10 and S11 the left and right panels association with transverse and vertical components respectively.

Comment #6: L250 and 259-260 - Why can the authors identify the dominant mode conversions from the dispersion curves? Fig. S9 shows the fundamental and higher modes of the Rayleigh waves, and it seems that both modes are possibly converted to Love wave. Also, it is hard to distinguish those modes in the records section in Fig. 5a. Please explain more details on the mode conversions in the text.

Suggestion) Is it possible to identify the fundamental and higher modes of the Rayleigh and Love waves in the record section in Figs. 4 and 5? Those in Fig. S13 are very clear, but it is nice if the authors specify them in those figures.

In the case of the source model S3, we previously stated: *“In this configuration, fundamental Love waves are generated mainly from higher Rayleigh modes which dominate the wavefield in the sediments (Suppl. Fig. S9).”* We agree with the reviewer that according to Fig. S9 both fundamental and higher modes are present at similar amplitudes and that therefore the sentence is not appropriate. We have reworked the sentence now line 260: *“In this configuration, fundamental mode Love waves and first mode Rayleigh waves are dominating in the deep water. Those phases result from conversions of fundamental and higher modes Rayleigh waves excited in the sediments (Suppl. Fig. S9) and interacting with the edge of the sedimentary basin.”*

In the case of source of model 4, for clarification, we have also rewritten part of the section, now line 270: *“For this model, the Rayleigh wavefield is dominated by higher modes in the sedimentary basin and deep water (mode characterization shown in Supplementary Figure S12), whose conversions at the shelf break result in both Rayleigh and Love fundamental modes. In contrast, in a model configuration where the Rayleigh fundamental mode is clearer in the deep water with limited higher modes (Suppl. Fig. S13), the pseudo-Rayleigh waves fundamental mode seems to convert to apparent Rayleigh waves seen on the transverse component.”*

We thought about specifying the modes on the section records of Fig. 4 and 5. However, as seen on Figure S9 and S12, for a sediment velocity of $V_p=2.5$ km/s, modes 0, 1 and 2 are quite tangled at the periods of interest.

Comment #7: L275 - station located on the shelf -> station (st15 in Fig. 5) located on the shelf
We have actually reworked that part to clearly state that we are looking at *“two separate stations located on the shelf”*. Now line 286

Comment #8: Fig. 6 - I do not understand the right panels. What are “Bathy.” and “Sed.”?
Please explain.

We agree it is a bit confusing, we were actually differentiating here the models with changing water column thickness from models with changing sedimentary basin thickness. After reworking Figure 6 to make it clearer, we decided this differentiation was not essential and we merged both “Bathy.” and “Sed.” plots into a signal one for more clarity.

Comment #9: L368 - gird -> grid?

We have made the correction. Now line 381

Comment #10: L416 - Although the authors state “quite consistent”, it seems that T/Z and L/R in the synthetic calculations are lower than those in the observations. It is better to explain the consistency more details, or remove “quite consistent” from the text and explain possible reasons for the difference between the calculations and the observations.

We agree with the reviewer suggestion and have decided to removed “quite consistent” as it is a confusing term. We have reworked the sentence, now line 431: “*The L/R ratios estimated from 3D simulation results are slightly lower than for the real data estimates over a year (Fig. 9), but both synthetic and real ratio distributions overlap well.*”

Possible reasons explaining the differences between observed and synthetic estimates are described at line 440.

Comment #11: L542 - DA and GA should be explained in more appropriate part, because these abbreviations are used in L122 and L515-516.

We agree and now define the abbreviations where they are first mentioned in the text. Now line 126.

Comment #12: L554-583 - I do not understand the details in this part. The following contents are necessary to understand this part.

We have reworked the section (now line 572) and hope it is now clearer. Changes from the original version can be seen in the manuscript version with tracked changes.

What kind of the observed data did the authors use in this part to obtain vertical red lines in Fig. S27?

Now explained line 583: “*The distribution of data points (here the back-azimuth) can be illustrated through a single variable plot such as a rug plot (Supp. Fig. S27a), which plots the back-azimuth as vertical lines along the x-axis.*”

Is $f(x)$ the probability density map, and is x the location?

$f(x)$ is the relative distribution as a function of the location which is normalized to give a probability density (i.e. it sums to 1). However, thanks to the reviewer comment we realized that this equation and associated sentence did not make sense here, as we are actually looking at relative distributions and not probability distributions on Figure 1 and 8. We have modified the section and have changed the labels on Figure 1 and 8 accordingly.

Did the authors use this technique to make Fig. 1 using the data in Figs. S1-S3?

Yes, we have now added a statement line 606 to make this clearer: “*This approach is used on all components and two period bands (3-5s and 5-8s). The source distribution maps of Figure 1 and Figure 8 are generated using the back-azimuth data of Figures S1-S3 and synthetic back-azimuth data of Figures S21—S23 respectively.*”

I like your record sections for wave propagations, which allowed me to compare the three component amplitudes easily. Thanks so much for your hard works.

REVIEWERS' COMMENTS

Reviewer #3 (Remarks to the Author):

Thank you very much for revising the manuscript. I am satisfied with this version, and (to me) the manuscript is now ready for publication. Also, it is better to consider the followings before the publication.

- In the Methods section, 'baz' and 'BAZ' are contaminated, so it is better to unify to either.
- The abbreviation 'hrfk' may be changed to HRFK.

REVIEWER COMMENTS

Reviewer #3

Thank you very much for revising the manuscript. I am satisfied with this version, and (to me) the manuscript is now ready for publication. Also, it is better to consider the followings before the publication.

–In the Methods section, ‘baz’ and ‘BAZ’ are contaminated, so it is better to unify to either.
We have now modified baz to BAZ, line 591.

–The abbreviation ‘hrfk’ may be changed to HRFK.
We now have changed hrfk to HRFK, line 546, 548 and 582.